# Recent Developments in Dielectric Barrier Discharge Plasma-Assisted Catalytic Dry Reforming of Methane over Ni-Based Catalysts

**Xingyuan Gao** [1,2,†], **Ziting Lin** [1,†], **Tingting Li** [1], **Liuting Huang** [1], **Jinmiao Zhang** [1], **Saeed Askari** [3], **Nikita Dewangan** [3], **Ashok Jangam** [3] and **Sibudjing Kawi** [3,*]

1   Department of Chemistry, Guangdong University of Education, Guangzhou 510303, China; gaoxingyuan@gdei.edu.cn (X.G.); lziting@gdei.edu.cn (Z.L.); ltingting15@gdei.edu.cn (T.L.); huangliuting@gdei.edu.cn (L.H.); zhangjinmiao@gdei.edu.cn (J.Z.)
2   Engineering Technology Development Center of Advanced Materials & Energy Saving, Emission Reduction in Guangdong Colleges and Universities, Guangzhou 510303, China
3   Department of Chemical and Biomolecular Engineering, National University of Singapore, Singapore 117585, Singapore; e0554257@u.nus.edu (S.A.); dnikita@u.nus.edu (N.D.); chejang@nus.edu.sg (A.J.)
*   Correspondence: chekawis@nus.edu.sg; Tel.: +65-6516-6312
†   The authors contribute equally to this review.

**Abstract:** The greenhouse effect is leading to global warming and destruction of the ecological environment. The conversion of carbon dioxide and methane greenhouse gases into valuable substances has attracted scientists' attentions. Dry reforming of methane (DRM) alleviates environmental problems and converts $CO_2$ and $CH_4$ into valuable chemical substances; however, due to the high energy input to break the strong chemical bonds in $CO_2$ and $CH_4$, non-thermal plasma (NTP) catalyzed DRM has been promising in activating $CO_2$ at ambient conditions, thus greatly lowering the energy input; moreover, the synergistic effect of the catalyst and plasma improves the reaction efficiency. In this review, the recent developments of catalytic DRM in a dielectric barrier discharge (DBD) plasma reactor on Ni-based catalysts are summarized, including the concept, characteristics, generation, and types of NTP used for catalytic DRM and corresponding mechanisms, the synergy and performance of Ni-based catalysts with DBD plasma, the design of DBD reactor and process parameter optimization, and finally current challenges and future prospects are provided.

**Keywords:** plasma; dry reforming of methane (DRM); dielectric barrier discharge (DBD); Ni-based catalyst

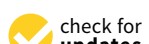

## 1. Introduction

With industrial development and social progress, a large amount of fossil fuels have been burned, energy consumption in various places has increased sharply, carbon dioxide emissions have increased greatly, and the greenhouse effect has become a serious issue, so reducing the negative impact of climate change and slowing down global energy consumption are of vital importance. At the same time, greenhouse gases ($CH_4$ and $CO_2$) are also deemed as the raw feed for production of value-added chemicals [1,2]. Therefore, dry reforming of methane (DRM) has been put into industrial production and has attracted widespread attention because of its dual benefits of environmental protection and resource utilization [3–7]. The product is generally synthesis gas or other valuable chemical substances, such as hydrogen and carbon monoxide (syngas components), ammonia, methanol, acetic acid, synthetic gasoline, etc. The main reaction equation is:

$$CH_4 + CO_2 = 2CO + 2H_2, \Delta H_{298K} = 247 \text{ kJ/mol}$$

because of the high stability of methane and carbon dioxide molecules. Studies have shown that DRM has the following advantages: (1) Wide sources of raw materials are

utilized to turn waste into treasure, thereby reducing atmospheric pollution; (2) compared to wet reforming and partial oxidation reforming of methane, DRM can save nearly half of methane; (3) the ratio of $H_2$ to CO is close to 1, appropriate for oxo reaction (hydroformylation reaction in the presence of Co or Rh where olefins reacts with CO and $H_2$ to produce aldehydes) and FTS reaction (also known as Fischer-Tropsch Synthesis, a process where CO and $H_2$ react to form olefins and other valuable products); (4) the reaction has a large reaction heat, which can be used as energy storage and medium transmission, such as solar energy storage [1,8–17].

However, in recent years, the traditional DRM method has gradually shown some limitations, such as high energy consumption and catalyst deactivation at high operation temperatures [9,18]. In this case, the emergence of plasma technology overcomes the high energy input to activate methane and carbon dioxide molecules. The high-energy electrons produced by plasma can initiate chemical reactions at room temperature, providing a new way for reforming reactions. Traditional methane conversion underwent a two-step catalytic process, usually conducted under intensive conditions. In contrast, non-equilibrium plasma-assisted DRM where electrons possess a higher energy owns a much lower reaction temperature than that of the traditional reforming process. In other words, the non-equilibrium plasma has the potential to generate active species in situ at low temperatures to accelerate the reaction process [8,19].

Non-thermal plasma technology (NTP) has attracted more and more attention due to its simple equipment, easy operation, high conversion rate; moreover, NTP saves the energy and materials and benefits the environmental protection. Current studies have shown that a variety of NTP structures have been tested for DRM reactions, such as gliding arc discharge, dielectric barrier discharge (DBD), microwave discharge, corona discharge,, etc. Among them, DBD has drawn special attentions because of its high density of electrons and production of highly active species. Scientists have conducted a lot of research on related areas [8,10,20–22].

Despite many advantages of DBD in DRM, the reaction has a lower activity and serious coke formation without a catalyst. Integrating NTP with the catalyst, a synergistic effect is observed to increase the conversions of reactants and the selectivity or yield of the targeted product due to the specific excitation formed in the plasma with sufficient energy. In detail, the electric field of plasma is enhanced due to charge accumulation and a polarization effect caused by the roughness of the catalysts. Additionally, a higher interaction between the catalyst and active species causes a higher conductivity, which improves the magnitude of the electric field. Meanwhile, the catalyst facilitates the adsorption and prolongs the contact time of reactants, leading to a higher activation and conversion. On the other hand, the highly reactive species generated by plasma cause the structure change and surface faceting of the catalyst, promoting the charge deposition and hotspot formation, which activates the metals and reactants. The combination of plasma and catalysts can prolong the lifetime of active species and lower the activation barrier [8]. Among the catalysts for plasma-catalyzed DRM, noble metals show good catalytic performances. As seen from previous studies, precious metals show strong resistance to coke formation and exhibit high catalytic activity in thermal and plasma DRM. The precious metals commonly used in plasma DRM include Pt, Pd, Rh, and Ru [8]. For example, Pt nanoparticles were impregnated in the UiO-67 MOF structure, accelerating the reactant dissociation and enhancing $H_2$ yield. Moreover, due to the dehydrogenation of hydrocarbons on Pt nanoparticles, the selectivity towards light hydrocarbons was reduced by 30%. Assisted by plasma, the surface reactions were intensified, increasing the energy efficiency by 11%. Furthermore, excellent stability was shown by the constant conversions during four cycles of plasma on-off [23]. However, due to their scarcity, high price, and easy sintering at high temperatures, they are not suitable for large-scale industrial applications [24,25]. Relatively speaking, transition metals such as Co, Mn, Fe, Cu, and Ni are also used for plasmonic catalytic DRM reactions. Zeng et al. compared the performance of different $\gamma$-$Al_2O_3$ supported metal catalysts M/$\gamma$-$Al_2O_3$ (M = Ni, Co, Cu, and Mn) in the plasma assisted DRM. They found out that the

combination of plasma with $Ni/\gamma-Al_2O_3$ and $Mn/\gamma-Al_2O_3$ catalysts significantly increased $CH_4$ conversion [26,27]. Other studies have shown that among various transition metals (Ni, Co, and Fe) used to catalyze DRM, Ni based catalysts presented good activities and economic feasibility [28–31], currently considered one of the most promising catalysts in DRM reactions due to the high catalytic activity and low cost [32–38].

However, the recent progress of DRM catalyzed by DBD plasma on Ni-based catalysts is rarely summarized. Therefore, this article focuses on the application of DBD-Ni catalysts integrated system for DRM reaction, discusses the synergy between Nickel-based catalysts and DBD reactor, introduces the reactor design and process parameter optimization, and finally provides the current challenges and prospects for the future.

## 2. Overview of Non-Thermal Plasma

Plasma takes up around 99% of the universe substances, which is a term used to describe an ionized gas. It is a macroscopic appearance composed of free electrons, ions, and neutral particles (in which positive and negative charges are equal), forming an electrically neutral non-condensing system [39,40]. The plasma as a whole is not charged, but because it contains free charge carriers, it is conductive. In addition, it has strong chemical activity. Many chemically stable substances can be activated by plasma [2].

The classification of plasma is divided into thermal plasma (equilibrium plasma) and non-thermal plasma (non-equilibrium plasma) according to the temperature of heavy particles inside the plasma and the thermodynamic balance. In thermal plasma, the temperature of its electrons and heavy particles are approximately equal, between 5000 and 50,000 K; including arc discharge plasma and inductively coupled plasma, often used in solid waste incineration and arc welding [2,40]. The electron temperature in NTP is much higher than that of heavy particles. Therefore, NTP technology has a wide range of applications such as pollutant removal, nano-material synthesis, material surface modification, and fuel reforming [24,26].

### 2.1. Characteristics of Non-Thermal Plasma (NTP)

NTP has non-equilibrium characteristics, that is, it can simultaneously have higher electron energy and lower ion and gas temperature. Electrons with sufficiently high energy can activate reactant molecules by dissociation and ionization during collisions; also, the reaction system can be kept near room temperature, which can reduce the energy consumption. Therefore, non-thermal plasma has a wide range of applications [25,39].

The energy in the non-thermal plasma is mainly used to generate highly reactive species (such as methane and carbon dioxide). The non-equilibrium characteristics can overcome thermodynamic obstacles in chemical reactions (such as dry reforming), allowing the reaction to proceed at room temperature and pressure. Besides, the on-off switch time of non-thermal plasma is so short that the excess energy generated by the fluctuation of the power grid can be used to achieve the stability of the power grid and facilitate the control of the reaction [35]. Therefore, non-thermal plasma technology is promising for methane dry reforming.

### 2.2. Non-Thermal Plasma Generation

Plasma can be generated from neutral gas by thermal excitation. When the gas is strongly heated up to a certain temperature, usually thousands of kelvins, the gas molecules form plasma due to their enough energy for spontaneous dissociation, excitation and ionization. However, due to technical issues and high energy input, this plasma generation method is not popular [2].

In contract, the widely accepted and facile NTP generation method is by means of electricity. Capacitively coupled plasma is a typical example, where a large potential difference is exerted between two electrodes. Due to the electrical discharge generated by the electric field, the gas between the electrodes is transformed into plasma. Considering the thousands of volts voltage and short distance between electrodes, the intensity of

the electric field in gas is high enough to accelerate the electrons from one electrode to another [2]. The current common non-thermal plasma mainly includes: Dielectric barrier discharge, corona discharge, gliding arc discharge, glow discharge, microwave discharge, and radio frequency discharge.

### 2.3. Mechanism of DRM Catalyzed by DBD Plasma

DBD is defined in the scenario where the dielectric is placed between two electrodes, and the discharge space is filled with an insulating medium. Due to the existence of the medium, the growth of the discharge current is limited, so as to avoid the complete breakdown of the gas and the formation of sparks or Arc [26]. When electricity is applied, plasma can be generated via the charge accumulation on the dielectric material with the short lifetime streamer, which will stop the discharge at one point, thus driving the discharge to happen at another point on the surface [41]. Moreover, DBD is a technology to generate stable plasma in a large scale at a low temperature and to produce electrons with a high energy to activate and dissociate greenhouse gas molecules such as $CH_4$ and $CO_2$, which are usually chemically stable due to the high bonding energies. On the other hand, dielectric barrier discharge can generate stable and uniform atmospheric pressure plasma under atmospheric pressure or higher. After optimization, this type of discharge has a broad development prospect in the industry.

DBD plasma is a means to effectively activate molecules. First of all, it can often activate high-stability methane molecules under milder conditions, usually at a pressure of 104–106 Pa and a frequency of 50 Hz [42]. Secondly, the simple design and easy-to-operate characteristics of the DBD reactor facilitates the miniaturization or expansion with high portability [39]. At the same time, DBD plasma shows certain advantages to DRM because of its low energy input, production of various active species, and low installation cost [18,43]. Therefore, this technology has been extensively studied in DRM [44].

Despite the advantages of DBD-catalytic assisted DRM over the single DRM, the low energy utilization efficiency limits its application [10,21,26]. To solve the problem, the combination of plasma technology and catalysts can improve the reactivity and energy utilization efficiency. It is known that the design of the plasma reactor should enable the plasma species optimally transported to the catalyst surface. Due to the high temperature of the catalyst, it is not simple to integrate the catalyst in the warm plasma; therefore, for plasma catalysis, DBD plasma as the representative NTP is more suitable [10,21].

### 2.4. Synergy of Non-Thermal Plasma and Catalyst

Although DRM shows a better performance in dielectric barrier plasma reactors, the traditional method to convert a large amount of $CO_2$ or $CH_4$ has many limitations. For example, other possible reactions, such as methane cracking, lead to reduced energy efficiency and residual carbon deposition [21]. In addition, the DRM process is highly endothermic with a long duration and high energy consumption. Therefore, the advantages of the combination of plasma and catalyst are exploited. Compared with the catalyst-free DRM, plasma-assisted catalytic DRM exhibited a higher reaction rate and strong resistance to coking due to the synergistic effect [45,46]. In detail, adding the catalyst to the discharge area of the DBD plasma can affect the efficiency of the system. Charge accumulation and polarization effect caused by the various shape and surface properties of catalysts enhance the electric field of plasma [8]. In other words, The plasma-catalyst interface increases the likelihood of reactant collisions and enhances surface modification and electric fields. These factors are beneficial to the treatment of gas in the DBD plasma DRM. On the other hand, plasma physically and chemically affects the surface area and interfacial chemistry of the catalyst, which in turn enhances DRM activity and improves product distribution [47].

The main factors affecting the plasma performance (measured in terms of output voltage and current intensity) are electric field enhancement and discharge. The roughness of the catalyst has a certain impact on it, due to the enhanced polarization effect and accumulation of charge in a more dispersed and smaller particle with high surface area

and high density of metal sites [48]. The combination of the catalyst and the DBD plasma increases the reactivity of most reactants by keeping them in excited state and lowering the activation energy [8,49]. In a word, the synergy between catalysts and DBD plasma can be summarized based on Figure 1 as below: First, the carbon deposition is alleviated; second, the conversions, selectivity, and stability are enhanced; third, catalyst reusability is improved; fourth, energy efficiency is increased [8].

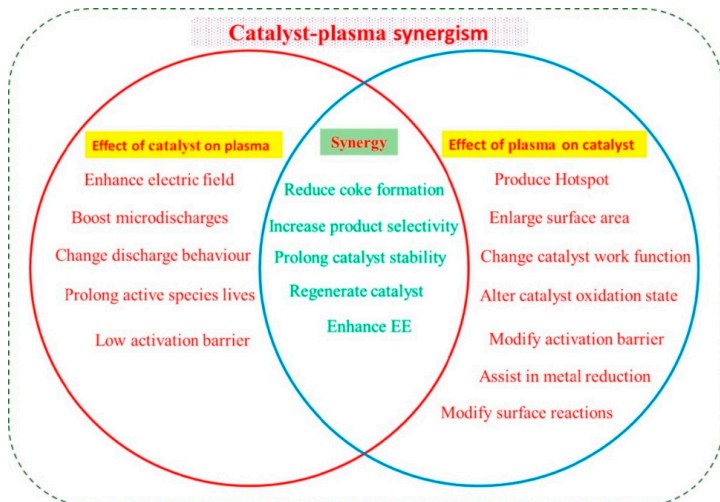

**Figure 1.** Synergy of the catalytic dielectric barrier discharge (DBD) plasma system. Reproduced with permission from [8], Copyright 2019, Elsevier.

In recent years, many studies have been focused on the application of a plasma synergistic catalyst to the DRM process [9,26,50]. A series of Co, Ru, Mn, and Pt-based catalysts have been published and most of them can increase the yield of products. Among them, Ni has become a hotspot due to the same stability, high activity, and high availability for DRM reactions as noble metals but with a lower cost [51]. For example, Wang et al. compared two DRM processes: Plasma-single and plasma-catalysis [26]. The results show that when Ni/C catalyst is combined with plasma, the catalytic activity was enhanced and the reaction conversion rate increased. Pietruszka et al. found that by combining plasma and Ni/$\gamma$-Al$_2$O$_3$, conversions of both CH$_4$ and CO$_2$ were increased due to the discharge heating of the catalyst [24]. Kim et al. observed that with Ni/Al$_2$O$_3$ catalyst, conversion of CH$_4$ was significantly higher than that of DBD without catalyst [45,50,52,53]. Therefore, this article summarizes the latest progress of Ni-based catalyst assisted by DBD plasma for DRM.

## 3. Ni-Based Catalyst Assisted with DBD Plasma for DRM

Due to the high availability and low cost of nickel-based catalysts, it is widely considered in plasma DRM. The common supports used in Ni-based catalysts are Al$_2$O$_3$, La$_2$O$_3$, SiO$_2$, AC, ZrO$_2$, and multi-element supports [8]. Studies have shown that different supports used in Ni-based catalysts exhibit different performances. For example, coking and metal sintering can be reduced in inert supports such as silica, thus improving the catalytic activity because CH$_4$ and CO$_2$ are both activated by metals with the help of plasma [8]. Besides, Ni-impregnated alumina (acid) and magnesium oxide (alkaline) catalysts have been widely used in plasma DRM, and their catalytic property has been improved by the dual-function approach, where CH$_4$ is activated by metal and CO$_2$ is activated by alkaline supports [54]. Specifically, alumina's high specific area, high thermal resistance, and good dielectric properties make it the preferred support for plasma catalysis [53]. In addition, Ni/ZrO$_2$ decomposed by plasma has a higher specific surface area, which generates more surface Ni active centers [55] (Table 1). Furthermore, activated carbon (AC) has the advan-

tages of low cost, large specific surface area, good stability, chemical inertness, and has also been applied to DRM [56].

### 3.1. Pure Ni Catalysts with Different Supports

#### 3.1.1. Ni/SiO$_2$

In the traditional DRM catalysis process, the sintering of nickel catalyst under high temperature reaction (generally above 500 °C) is one of its main disadvantages. The aggregation of nickel particles leads to the loss of active surface, thereby reducing activity and selectivity [57]. On one hand, DBD plasma and Ni/SiO$_2$ catalyst show a synergy, which inhibited coke formation. On the other hand, due to the higher energy of electrons in DBD than the bond dissociation energy of nickel, metallic Ni crystal grains are reduced to smaller crystal grains on the Ni/SiO$_2$ catalyst [58]. In the strong electric field of the DBD, the Ni-support interaction is affected by high-energy electrons, resulting in the dispersion of Ni particles on the catalyst support. In addition, DRM catalyzed by Ni/SiO$_2$ occurs at the low temperature and ambient pressure of non-thermal plasma, which may prevent nickel from sintering on the catalyst. As shown in Figure 2, a broader peak shape was presented after the reaction in 2b, suggesting a smaller size for the spent Ni catalyst due to the assistance of non-thermal plasma [59] (Table 1).

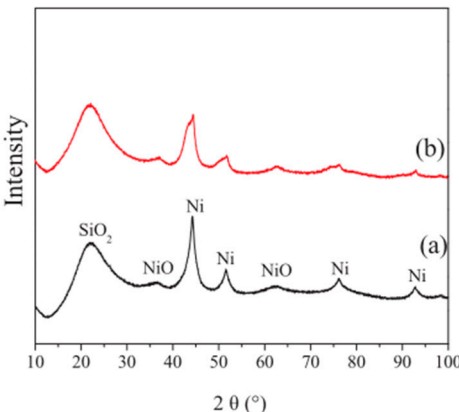

**Figure 2.** XRD pattern of Ni/SiO$_2$ catalyst sample. (**a**) Fresh catalyst. (**b**) Spent catalyst after 5 h test. Reproduced with permission from [59], Copyright 2015, Elsevier.

In addition, the Ni/SiO$_2$ catalyst can increase the CO selectivity, because the reaction between carbon-containing intermediates and oxygen radicals are catalyzed in the plasma. However, in the Ni/SiO$_2$ catalyst catalysis process, the conversion rate of CO$_2$ and CH$_4$ will be reduced, which can be attributed to the reverse reaction on the catalyst [59].

#### 3.1.2. Ni/Al$_2$O$_3$

Due to its high thermal resistance, high specific area, and good dielectric properties, alumina was once known as the preferred carrier for plasma catalysis. Recent studies have shown that among Ni/MgO, Ni/γ-Al$_2$O$_3$, Ni/TiO$_2$, and Ni/SiO$_2$, Ni/γ-Al$_2$O$_3$ catalyst performed the best [60]. Nickel-supported Al$_2$O$_3$ has been widely used and reported in NTP-catalyzed DRM [54]. Tu and Whitehead reported that a synergistic effect of Ni/Al$_2$O$_3$ with low-temperature plasma almost doubled the methane conversion and H$_2$ yield compared with DRM with plasma alone [37,61].

L. Brune and coworkers also performed DBD plasma-DRM reactions with Ni/γ-Al$_2$O$_3$ as fillers [21]. Results showed that porous alumina with a huge specific area captured CO2 molecules, which significantly increased its conversion. Additionally, with the same wall plug power, total flow rate, and operating frequency, porous γ-Al$_2$O$_3$ beads filled in the gaps of the DBD increased the CH$_4$ conversion rate by nearly 25% compared with the plasma alone [21,62].

To enhance $H_2/CO$ ratio in $Ni/Al_2O_3$ catalyst coupled with DBD reactor, Thitiporn et al. increased the $CH_4/CO_2$ feed ratio to 4 and a high $H_2/CO$ ratio of 1.5 was realized; moreover, with 5% Ni loaded on $Al_2O_3$, only 3.7 wt% carbon deposition was presented, which was 4.7 wt% without Ni, suggesting the synergy between Ni catalyst and DBD [63] (Table 1). However, even though $Ni/\gamma-Al_2O_3$ exhibits good catalytic performance, there is still a problem of carbon deposition. The deposited carbon covers the active center of the catalyst and causes metal particles to sinter, thus deactivating the catalyst and reducing the DRM performance [43,64].

### 3.1.3. $Ni/ZrO_2$

$ZrO_2$ has been studied as the packing materials in DBD plasma reactor to catalyze the DRM reaction. However, with $ZrO_2$ alone, the fraction of void decreased, reducing the current intensity and plasma generation. Moreover, surface discharges became dominant instead of filamentary microdischarges, negatively affecting the activation of reactant molecules [23].

In contrast, $Ni/ZrO_2$ exhibits an enhanced reactivity in DBD DRM reaction. The nickel precursor is decomposed under the DBD plasma to prepare a nickel/zirconia catalyst, possessing a high specific surface area and provides more surface Ni active centers for the reaction. During the early work, the application of $Ni/ZrO_2$ catalysts in dry reforming has been studied, and it is found that carbon deposition is significantly reduced, especially under low nickel loading, if the nickel microcrystals used are very small [65]. $ZrO_2$ contains two crystal phases, namely monoclinic ($m-ZrO_2$) and tetragonal ($t-ZrO_2$). In $ZrO_2$-P (precursor decomposed with plasma treatment in Ar atmosphere), the amount of $t-ZrO_2$ is twice that of $Ni/ZrO_2$-C (precursor calcined in air without plasma treatment), indicating the beneficial effect of NTP treatment on the formation of $t-ZrO_2$ [55]. Compared with $Ni/ZrO_2$-C, the agglomeration of nickel particles on the $Ni/ZrO_2$-P was much reduced, and the exposed surface lattice fringes of the Ni particles were clearly visible. In addition, more oxygen vacancies were generated on $Ni/ZrO_2$-P, which provided stronger alkalinity and promoted $CO_2$ adsorption and activation, enhancing the activity of $Ni/ZrO_2$ in DRM. Meanwhile, oxygen vacancies additionally drive the $CO_2$ reduction, resulting in enhanced activation of $CO_2$ and $Ni/ZrO_2$ DRM activity [56,66]. The methane and $CO_2$ conversions using plasma and catalyst were 53.57% and 60.81%, higher than 42.30% and 52.88% with plasma alone, indicating that the combined treatment of DBD plasma and $Ni/ZrO_2$ can provide stronger adsorption sites and improve the adsorption capacity of $CO_2$, further increasing the activity of $Ni/ZrO_2$ in DRM. Finally, by comparing the carbon deposition amount of the two catalysts after the DRM reaction, $Ni/ZrO_2$-P possessed lower coke formation (The carbon deposition amount of $Ni/ZrO_2$-P and $Ni/ZrO_2$-C is 33 wt% and 64 wt%, respectively) [55].

In summary, studies by Vakili and Hu et al. [23,55] have confirmed that DBD plasma decomposition benefits the formation of highly active Ni-based DRM catalysts. Through plasma decomposition, $Ni/ZrO_2$-P possessed a smaller nickel size and denser Ni (111) planes. At the same time, it has higher dispersion, more $t-ZrO_2$ and more oxygen vacancies, which helps $CO_2$ adsorption and activation. Owing to the synergistic effect of non-thermal plasma and $Ni/ZrO_2$, the catalytic activity and anti-coking ability for DRM are effectively enhanced [55,65].

### 3.1.4. Ni/AC

Carbon materials such as activated carbon (AC) are characterized with low cost, good stability, large specific surface area, and chemical inertness [56]. In addition, activated carbon is effective for oxygen reduction, controlling the surface chemistry and pore size distribution. In recent years, many studies have been launched on this field. For example, Wang et al. studied DRM with plasma alone, catalyst alone, and catalyst-assisted plasma to determine the synergy [60]. In the case of catalytic mode, since DRM is an endothermic reaction, a higher operating temperature is required to achieve a high balance of $CH_4$ and

$CO_2$ conversion. Considering the low temperature of 270 °C, the conversions were limited. With the plasma alone, the $CH_4$ and $CO_2$ conversion were 51.5% and 42.0%, respectively. Compared with the reforming in the catalytic-single or plasma-single mode, due to the synergistic effect of plasma and catalyst, extremely high conversions were obtained of 64.6% and 65.7% for methane [60]. Based on the characterization results, higher porosity and larger specific surface area enhanced the activity and stability, suggesting a good application potential of the plasma catalytic DRM.

### 3.2. Ni-Based Catalysts with Doping

Due to the plasma-catalyst synergy, DRM activities are significantly increased. However, doping with other metals or compounds can effectively resist carbon formation and optimize the product distribution. Hao and Yashima showed that compared with $Ni/Al_2O_3$ or $Rh/Al_2O_3$ catalysts, Rh promoted $Ni/Al_2O_3$ exhibited a better performance [67]. Moreover, metals such as Co, K, Mg, Mn, La, and Ce have been used as modifiers recently. Their combinations with metal Ni produce a higher reactivity for DRM [20,25,54,61,68–71].

### 3.2.1. Transition Metals

Transition metals have drawn interest as a doping agent to promote the performance of Ni catalysts for DBD-catalyzed DRM reaction, such as Mn [20] and Co [68]. The Ni-$Co/Al_2O_3$-$ZrO_2$ catalyst with an amorphous structure can improve the active phase dispersion and enhance the metal-support interaction, which is more conducive to the activation of reactants in DRM than the $Ni/Al_2O_3$ catalyst. Nader Rahemi et al. proved that the synergistic effect of Ni-$Co/Al_2O_3$-$ZrO_2$ and plasma can achieve higher $CH_4$ and $CO_2$ conversions and $H_2$ and CO yields [68] (Table 1). The Ni-$Co/Al_2O_3$-$ZrO_2$ nanocatalyst treated under 1000 V presented a uniform morphology, large surface area, and small particle size of 21.2 nm averagely, suggesting the great effect of plasma voltage on the crystallinity and size of NiO. In detail, the amorphous active phase of Ni-$Co/Al_2O_3$-$ZrO_2$ was easily attached to the crystal lattice of the support. The plasma improved the irregularity of the structure by generating kinks, vacancies, and other structural defects, causing the crystal grains to lose the lattice arrangement, thereby improving the active phase dispersion and strengthening the metal-support interaction. In the meantime, the fluid state time of the catalyst showed that the Ni-$Co/Al_2O_3$-$ZrO_2$ catalyst combined with plasma improved $CH_4$ and $CO_2$ conversions and the $H_2/CO$ ratio compared with the traditional DRM [68].

Typical results suggest excellent activity and syngas yield under plasma conditions in Ni-$Co/Al_2O_3$-$ZrO_2$, however, the anti-carbon deposition and product selectivity of $Al_2O_3$-$ZrO_2$ catalyst can be improved. In this case, because Mn positively affects surface modification and activation of the catalyst, the emergence of Ni-$Mn/Al_2O_3$ catalyst causes widespread attentions. In Xintu's work, $Ni/Al_2O_3$ and Ni-$Mn/Al_2O_3$ catalysts were compared with plasma discharge [20] (Table 1). The experimental results showed that compared with plasma alone, catalytic DBD exhibited a higher conversion and syngas ratio. In detail, the $Ni/Al_2O_3$ catalyst possessed a strong affinity for $CH_4$, which is conducive to CO disproportionation, resulting in coke formation. Conversely, the Ni-Mn catalyst delivered a high conversion and stable activity. Under similar conditions, the amount of carbon produced on the surface of the $Ni/Al_2O_3$ and Ni-$Mn/Al_2O_3$ catalysts was 4.3 mg and 1.6 mg, respectively, which confirmed that the bimetallic catalyst Ni-$Mn/Al_2O_3$ was better than $Ni/Al_2O_3$. Moreover, a higher energy efficiency in DBD plasma-DRM was presented that when the discharge power was 1.0 W, the maximum energy efficiency of the bimetallic catalyst filled DBD was 2.76 mmol/kJ [20].

In general, DBD plasma reactor is well applied in DRM to produce hydrogen/syngas. Adding a catalyst to the DBD reactor can increase the conversions of reactants. The catalytic activity for $CH_4$ conversion is in order: Ni-$Mn/Al_2O_3$ > $Ni/Al_2O_3$ > Plasma alone. The best activity of the DBD filled with bimetallic catalyst is related to its resistance to carbon formation [20,72].

### 3.2.2. Alkali and Alkaline Earth Metals

In addition to transition metals, the addition of promoters such as alkali and alkaline earth metals (Mg and K [60,61]) can also inhibit the carbon formation, improve the dispersion of Ni metal, prevent metal sintering, strengthen the metal-support interaction thus enhancing the catalytic performance. One of the reasons is possibly their addition increases the alkalinity of the catalyst, thereby promoting the activation of $CO_2$ and $CH_4$ [31,65,73]. Recently, DeBeck and his colleagues discovered that hydrotalcite-derived catalysts (Ni-Mg/$Al_2O_3$) inhibited the Ni sintering and enhanced $CH_4$ conversion [74]. Tao et al. also doped Mg into the Ni/$Al_2O_3$ to enhance the $CO_2$ adsorption with the surface hydroxyl group, thus reducing the coke formation [75]. Sengupta et al. found that adding 5wt.% MgO to the Ni/$Al_2O_3$ catalyst can significantly reduce the carbon formation on the spent catalyst from 24.5 wt% to 14.4 wt% [76]. Ozkara-Aydinoglu et al. successfully reduced the carbon deposition from 4% to 2.6% when using Mg promoter in Co/$ZrO_2$ catalyst [77].

The mechanism can be summarized as below: the addition of Mg promoter can increase the number of strongly basic sites on the catalyst, thus significantly improving $CO_2$ adsorption. Coupled with DBD plasma, Ni-Mg/$Al_2O_3$ catalyst activated $CO_2$, but inhibited $CH_4$ conversion, thus reducing the carbon deposits in the product. On the TPR spectrum of the Ni-Mg/$Al_2O_3$ catalyst as shown in Figure 3, compared with the Ni/$Al_2O_3$ catalyst, the reduction temperature of NiO increases and the peak intensity related to the reduction of NiO decreases, suggesting that due to the plasma catalytic system and Mg promoters, the interaction between NiO and $Al_2O_3$ was greatly enhanced [61] (Table 1).

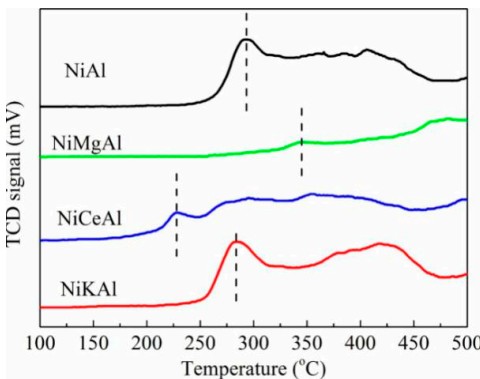

**Figure 3.** $H_2$-TPR spectrum of fresh catalyst. Reproduced with permission from [61], Copyright 2018, Elsevier.

Different from Mg, the addition of K the Ni-K/$Al_2O_3$ catalyst increases the conversions of the reactants and the yield of the products, and the energy efficiency of the plasma process is also greatly improved, which highlights the synergy between the plasma and the catalyst [60,61]. The interaction between DBD plasma and catalyst is the main driving force. Compared with plasma alone, integrating plasma with Ni-K/$Al_2O_3$ catalyst greatly increased $CH_4$ and $CO_2$ conversion to 31.6% and 22.8%, respectively. Moreover, $H_2$ selectivity was enhanced to 43.3% [61]. This is because the presence of K can increase the basicity of the DBD plasma catalytic system, which is beneficial to the conversion of carbon dioxide. Additionally, based on the TPR profiles, the reduction temperature of NiO in Ni-K/$Al_2O_3$ is significantly lowered, suggesting the formation of abundant active sites. However, compared with the Ni/$Al_2O_3$ catalyst, K doping increased the carbon deposition on the catalyst, consistent with the higher $CH_4$ conversions in the plasma catalytic system [60,61].

### 3.2.3. Rare Earth Metals

In the plasma-catalyzed DRM reaction, the Ni-K/$Al_2O_3$ catalyst is used to achieve a good yield of $H_2$ and CO. Similarly, rare earth metals such as La and Ce can act as doping agents to enhance the catalytic performance of Ni-based catalysts in DBD reactor for DRM [54,71]. Ce has been widely applied to improve the stability of DRM catalysts

by inhibiting carbon formation and participating in the gasification of carbon deposits that have already formed [70,78–80]. For example, Ni-Ce/$Al_2O_3$ catalyst can effectively enhance the reactivity with the combination of plasma [70]. Wang et al. studied the catalytic DRM on NiO-$CeO_2$-$Al_2O_3$ catalyst and found that by increasing the ratio Ce/Al in Ni-Ce/$Al_2O_3$ from 0 to 1:50 resulted in higher conversions of $CH_4$ and $CO_2$. The presence of Ce further improves the catalytic performance of DRM under plasma conditions since $CeO_2$ has a higher dielectric constant, thus exhibiting a better dielectric performance [79]. Moreover, compared with the Ni/$Al_2O_3$ catalyst, Ce addition reduced the $CO_2$ desorption temperature by 20 °C, indicating that the combination of DBD plasma and the catalyst can weaken the $CO_2$ desorption at strong basic sites owing to the enhanced activation of $CO_2$ on $CeO_2$ species [61,70].

In addition to enhancing the catalytic performance, combination of Ni-Ce catalysts and DBD can prevent the coke formation. Tao et al. found that Ce addition would bring oxygen vacancies and promote the adsorption of $CO_2$, thus facilitating the coke removal. Moreover, the high specific surface area of the catalyst changed the filament discharge alone to filament/surface discharge mode, increasing the discharge area [75] (Table 1). Besides, in the DBD-treated Ni/$SiO_2$ catalyst, the interfacial interaction was obviously improved, which promoted the activation and dissociation of methane. Additionally, the Ni/$CeO_2$-$SiO_2$ catalyst decomposed by DBD plasma showed smaller Ni particle size and fewer defect centers, beneficial to improve the activity, anti-coking, and anti-sintering performance in the DRM reaction.

To investigate the Ce doping effect in Ni/$SiO_2$ with DBD, Yan et al. compared Ni/$CeO_2$-$SiO_2$-C (no plasma treatment) and Ni/$CeO_2$-$SiO_2$-P (DBD plasma treatment) for DRM [71]. The experimental results showed how the interaction between Ni and $CeO_2$ improved the catalytic performance. At 700 °C, Ni/$CeO_2$-$SiO_2$-P delivered a higher activity and $H_2$/CO ratio than Ni/$CeO_2$-$SiO_2$-C. The conversions of $CO_2$ and $CH_4$ were 87.3% and 78.5%, higher than the latter (80.5% and 67.8%). At the same time, Ni/$CeO_2$-$SiO_2$-P is stable in long-term reaction, while Ni/$CeO_2$-$SiO_2$-C shows poor stability within 10h [71].

The main difference between the two catalysts lied in the amount of active oxygen in the interface structure between Ni and $CeO_2$. Ni/$CeO_2$-$SiO_2$-P has more active oxygen than Ni/$CeO_2$-$SiO_2$-C. As shown in Figure 4a,b, a higher ratio of $Ce^{3+}$/$Ce^{4+}$ of 0.24 was achieved in Ni/$CeO_2$-$SiO_2$-P than that of 0.18 in Ni/$CeO_2$-$SiO_2$-C, which suggested formation of more surface oxygen species. Additionally, Ni/$CeO_2$-$SiO_2$-P in Figure 4c presented a higher ratio of $O_\alpha$/$O_\beta$ where $O_\alpha$ referred to oxygen ions and $O_\beta$ belonged to the surface hydroxyl groups, indicating more active oxygen generated with the plasma treatment. Therefore, Ni/$CeO_2$-$SiO_2$-P exhibited a better catalytic performance and higher $H_2$/CO ratio at lower than 700 °C. Based on the kinetic study, reactants were more easily activated in Ni/$CeO_2$-$SiO_2$-P than Ni/$CeO_2$-$SiO_2$-C, which confirmed the higher reforming activity of the former. In general, Ni/$CeO_2$-$SiO_2$ with plasma exhibited a good synergistic effect in DRM [71].

In addition to $CeO_2$, another rare earth metal oxide, $La_2O_3$ is proven effective as a doping agent. For example, $MgAl_2O_4$ structure facilitates the interaction and dispersion of the metal and changes the acidity and alkalinity of the catalyst [54]. In order to further increase the $H_2$/CO ratio, studies have shown that adding $La_2O_3$ can inhibit the formation of water [54] (Table 1). Besides, $La_2O_3$ used for the dry reforming of methane shows significant resistance to carbon formation due to strong alkalinity and carbon gasification [81]. $La_2O_3$ inhibits carbon deposition and prolongs the stability of the catalyst because it can form an intermediate carbonate ($La_2O_2CO_3$), which further reacts with the carbon on the surface near nickel to form $CO_2$. The interaction of nickel and magnesium aluminoxane can also be enhanced by the assistance of $La_2O_3$ as an additive. Under plasma conditions, mixed support $La_2O_3$ and $MgAl_2O_4$ can increase the alkalinity of the catalyst, inhibit the aggregation of nickel particles, and reduce the generation of carbon [54,82] (Table 1). Owing to the merits, 10% nickel/$La_2O_3$-$MgAl_2O_4$ prepared in DBD plasma possessed the

highest $CH_4$ conversion (86%) and $CO_2$ conversion (84.5%), the highest $H_2$/CO ratio and the lowest carbon formation rate [54].

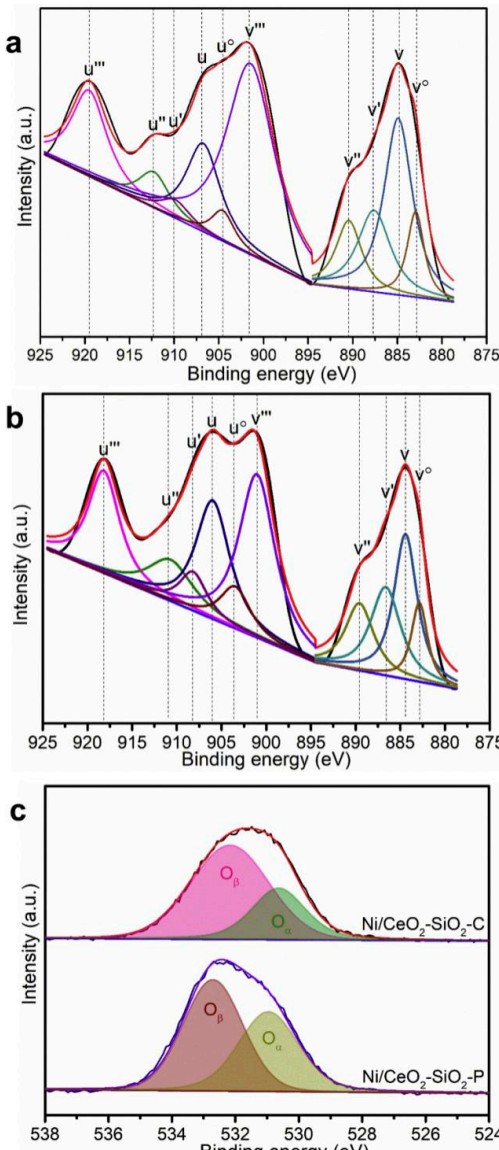

**Figure 4.** Ce 3d XPS spectra of (**a**) $NiO/CeO_2$-$SiO_2$-C and (**b**) $Ni/CeO_2$-$SiO_2$-P, and (**c**) O 1s XPS spectra of the two catalysts. Reproduced with permission from [71], Copyright 2019, Elsevier.

Finally, under similar experimental conditions, a comparative study of the synergistic effect of plasma and catalyst was carried out. The $CH_4$ and $CO_2$ conversions in the catalytic DBD plasma were 26% higher than those of the plasma alone. Due to the basic nature of the catalyst, the reactant gases were adsorbed on the surface of the catalyst, and more reactive substances diffused in the pores, allowing subsequent surface reactions to occur. When the reactant gas molecules collided with high-energy electrons, the electrons in the molecules were excited, dissociated, and ionized to produce active species such as O, H, methyl radicals, and finally CO and $H_2$. Therefore, the integration of $Ni/La_2O_3$-$MgAl_2O_4$ with the plasma facilitated the processing of the reactants and improved the efficiency of the DBD plasma reactor [54,83] (Table 1). In general, the preparation of the $Ni/La_2O_3$-$MgAl_2O_4$ improved the alkalinity of the catalyst, the Ni dispersion, the dielectric constant, and the interactions. Owing to these merits, the activation and chemical adsorption of reactants were enhanced.

**Table 1.** Catalytic DRM activities of Ni-based catalysts integrated with DBD plasma.

| Catalyst | Parameters | | | | Conversion(%) | | Selectivity(%) | | T (°C) | SIE * (J/mL) | Energy Efficiency (mmol/kJ) | Remarks | Ref. |
|---|---|---|---|---|---|---|---|---|---|---|---|---|---|
| | $CH_4/CO_2$ Ratio | Catalyst Loading (g) | Flow Rate (mL/min) | Power (W) | $CH_4$ | $CO_2$ | $H_2$ | CO | | | | | |
| $LaNiO_3@SiO_2$ | 1:1 | 0.2 | 40 | 150 | 88.31 | 77.76 | 83.65 | 92.43 | 200 | 225 | 0.17 | Integrated with DBD, $LaNiO_3@SiO_2$ shows improved conversions and selectivity. | [82] |
| $Ni/SiO_2$ | 1:1 | 0.2 | 50 | 86 | 26 | 16 | 47.4 | 52.9 | 110 | 103.2 | N.A. | The combination of DBD and $Ni/SiO_2$ catalyst enhance the activity of DRM due to the reaction between carbon-containing intermediates and oxygen radicals. | [59] |
| $Ni/Al_2O_3$ | 1:1 | 0.3 | 56 | 70 | 60 | 77 | 70–75 | 80 | 550 | 75 | 39% | The charge recombination on the $Ni/Al_2O_3$ catalyst surface will enhance the diffusion of carbon through the Ni catalyst and promote its oxidation by $CO_2$. | [49] |
| $Ni/Al_2O_3$ | 4:1 | 6.4 | 50 | 1600 | 8.3 | 7.6 | 69 | 20 | Ambient temperature | 4.6 eV/molecule | 4.5% | Ni addition enhanced the $H_2/CO$ ratio and reduced coke formation with the help of DBD. | [63] |
| $Ni/\gamma-Al_2O_3$ | 1:1 | 1.0 | 50 | 50 | 56.4 | 30.2 | 31 | 52.4 | <150 | 60 | 0.32 | The combination of plasma and $Ni/\gamma-Al_2O_3$ can increase the conversion rate of $CH_4$. | [27] |
| $Ni/La_2O_3-MgAl_2O_4$ | 1:1 | 0.5 | 20 | 100 | 86 | 84.5 | 50 | 49.5 | 350 | 300 | 0.13 | $La_2O_3$ inhibits the RWGS * reaction, improves $H_2$ selectivity and yield, and the formed intermediate carbonate ($La_2O_2CO_3$) inhibits carbon deposition. | [54] |
| $Ni/La_2O_3$ | 1:1 | 0.2 | 50 | 160 | 63 | 54 | 71 | 85 | 150 | 240 | 0.14 | $Ni/LaO_3$ nanoparticles show excellent thermal stability in the DBD plasma reactor. La contributes to the formation of intermediates, which are responsible for activating $CO_2$ and inhibiting carbon deposition. | [83] |
| $Ni/ZrO_2$ | 1:1 | 0.6 | 50 | 200 | 53.57 | 60.81 | 82 | 95 | 650 | 240 | N.A. | The $Ni/ZrO_2$ catalyst prepared by the DBD plasma decomposition method greatly improves its activity due to its high dispersion and increased oxygen vacancies. | [55] |
| $Ni-Co/Al_2O_3-ZrO_2$ | 1:1 | 0.3 | 40 | N.A. | 58 | 62 | 95 | 100 | 850 | N.A. | N.A. | The $Ni-Co/Al_2O_3-ZrO_2$ catalyst after plasma treatment shows high catalytic activity due to its narrow particle size distribution, large surface area and strong metal-support interaction. | [68] |
| $Ni-Mn/\gamma-Al_2O_3$ | 1:1 | 0.5 | 30 | 2.2 | 28.4 | 13.2 | 23.2 | 40.5 | N.A. | 4.2 | 2.76 | A higher activity and energy efficiency is achieved by the integrated plasma and Ni-Mn bimetallic catalyst system. | [20] |
| $Ni-Mg/Al_2O_3$ | 1.6:1 | 0.4 | 50 | 16 | 32 | 16 | 41.7 | 29.5 | 160 | 19.2 | 0.58 | K promoted catalyst shows the best performance and enhances the energy efficiency of plasma process because it contains more active sites. | [61] |
| $Ni-K/Al_2O_3$ | 1.6:1 | 0.4 | 50 | 16 | 34 | 23 | 43.3 | 31.3 | 160 | 19.2 | 0.67 | | |
| $Ni-Ce/Al_2O_3$ | 1.6:1 | 0.4 | 50 | 16 | 32 | 22 | 41.8 | 31.1 | 160 | 19.2 | 0.63 | | |
| NiMgAlCe | 4:6 | 0.45 | 90 | 48 | 36.1 | 22.5 | N.A. | N.A. | N.A. | 32 | N.A. | Mg and Ce promoted the $CO_2$ adsorption and increased the discharge area by partially tuning the filament discharge into surface discharge. | [75] |
| $Mg,Ce-Ni/\gamma-Al_2O_3$ | 1:1 | N.A. | 30 | 2.7 | 34.7 | 13 | 35 | 53.7 | N.A. | 5.4 | 1.97 | Mobile oxygen and surface basicity effectively removed coke during methane activation by Ni and DBD. | [80] |

* RWGS stands for the reverse water gas shift reaction; * SIE refers to specific input energy; N.A.: not available.

## 4. DBD Plasma Reactor Design

To achieve a high catalytic performance and energy efficiency, reactor designs including the configuration of the reactor, medium materials, and discharge volume play an essential role. In the following part, DBD plasma reactor designs will be illustrated in detail.

### 4.1. Configuration of DBD Plasma Reactor

Commonly used reactor configurations in DBD plasma include planar and cylindrical types. Typical planar DBDs reactor is composed of two electrodes (as shown in Figure 5), asymmetrically located on top and bottom sides of the dielectric barrier material, which limits the current and inhibits the sparks formation. Electric discharge is generated when an AC potential is exerted [24,26,63].

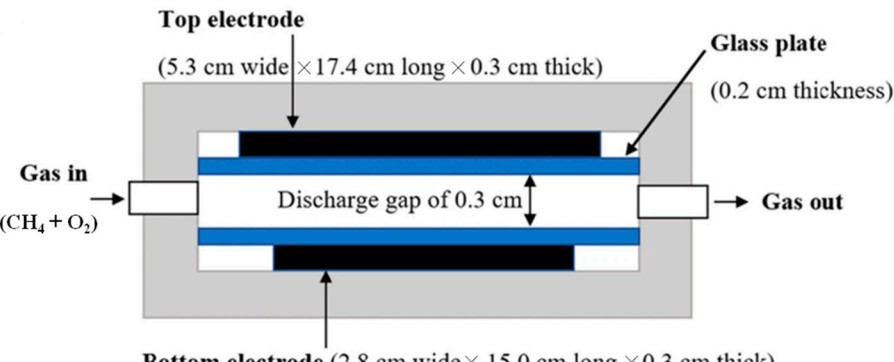

**Figure 5.** Two frosted glass plates are used to mix parallel plate DBD reactors. Reproduced with permission from [63], Copyright 2020, Springer.

Different from the planar DBD plasma reactor, as shown in Figure 6a, the cylindrical plasma reactor contains the following geometric parameters, which are discharge volume (VD, cm$^3$), the discharge length (DL, cm), discharge gap (D$_{gap}$, mm), and shape and material of the high-voltage electrode [8,41]. In the basic structure of the DBD reaction device, the high-voltage electrode is usually a rod-shaped material with high conductivity similar to steel and aluminum [84]. Generally, high-voltage electrode materials have three morphologies as shown in Figure 6b, including holes, porous, and rough structures [8]. The morphology of the high-voltage electrode is a very important factor in a DBD reaction device, usually determining whether the discharge is uniform. The uniform distribution of the high-voltage electrode can make the gas molecules and the energy-containing material fully interact, which improves the reaction rate of the DBD plasma in the reaction device. Moreover, researchers have studied the high-voltage electrode material and its surface morphology [22]. It is concluded that the surface conductivity can be increased by changing the high-voltage electrode surface morphology. The usual method is to deposit zinc oxide on the surface of the high-voltage electrode, thereby producing a more uniform surface and a stable plasma discharge [85]. On the other hand, the morphology will cause the discharge mechanism to be different, thus affecting the conversions of methane and carbon dioxide. With a porous electrode, a higher conversion will be obtained than that of a smooth high-voltage electrode. The reason is that the gas molecule fixation rate in the porous electrode is higher, thereby increasing the collisions of the active species DRM.

In addition to the rough morphology, uniformly coating the catalyst on the rough high-voltage electrode surface, as shown in Figure 6c, makes the catalyst active center more direct contact with the plasma [8]. Due to the robust discharge behavior and uniformity of the discharge, it can also produce reactive species. However, one drawback is carbon nanotube (CNT) growth and surface etching.

Besides the shape and morphology, the volume of the high-voltage electrode also greatly affects the discharge characteristics of the DBD plasma [8]. When reactant molecules

($CH_4$, $CO_2$) act in the reactor with DBD plasma, the volume of the high-voltage electrode affects the discharge volume, thereby influencing the conversion rate of $CH_4$ and $CO_2$.

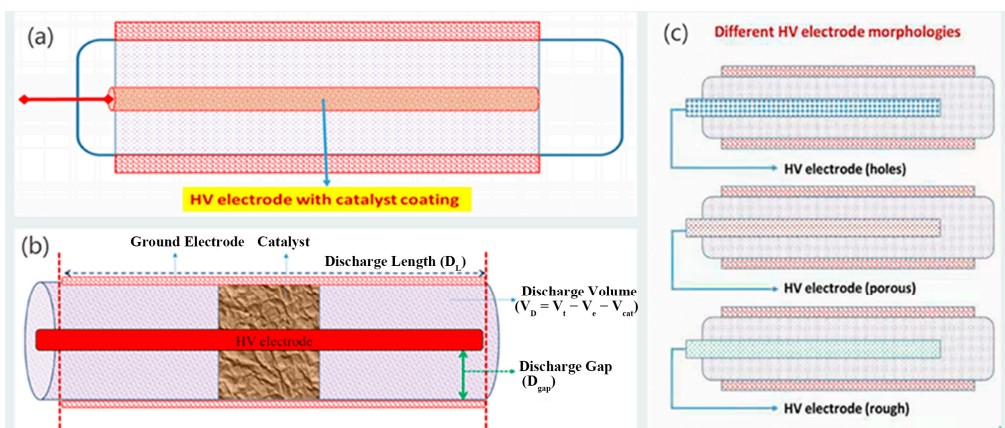

**Figure 6.** (**a**) The typical geometry of a DBD plasma reactor system; (**b**) the morphology of three different high-voltage electrode materials; (**c**) three morphologies of high-voltage electrode materials. Reproduced with permission from [8], Copyright 2019, Elsevier.

As one of the representative DBD reactors for plasma-assisted catalysis, packed-bed reactors with different designs are shown in Figure 7 [8]. Pre-packing mode is mostly applied in those temperature-controlled reactions. The catalysts needs to be activated first, followed by passing the reactants to the discharge zone, which may not be practical in DBD-DRM system. In contrast, in situ packing reactor takes advantages of the catalyst and plasma simultaneously. With a relatively stable temperature, the excited electrons promote the dissociation of reactant molecules. Few researchers focus on the post packing and fully packed DBD reactor in DRM reaction since they might be not effective as the in situ packing mode.

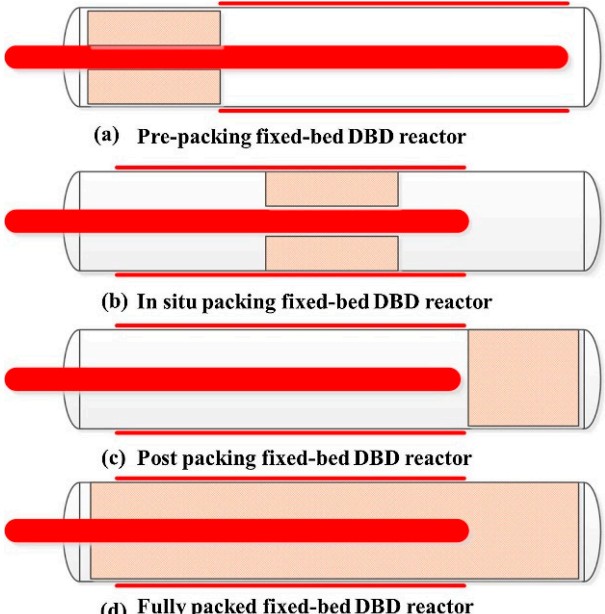

**Figure 7.** Different reactor configurations based on the packing material. (**a**) Pre-packing fixed-bed DBD reactor; (**b**) in situ packing fixed-bed DBD reactor; (**c**) post packing fixed-bed DBD reactor; (**d**) fully packed fixed-bed DBD reactor. Reproduced with permissionfrom [8], Copyright 2019, Elsevier.

### 4.2. Medium Material

The main function of the dielectric material in the DBD reactor is to separate the two electrodes of the reactor, namely the high-voltage and ground electrode; also, via adsorption of the discharge species, the discharge voltage is lowered. The dielectric material is mainly important for the mechanism of the discharge, because excitation, ionization, and dissociation are directly related to the dielectric constant in the plasma-induced reaction. The material with high dielectric constant can increase the temperature of the reactor by reducing the power consumption and thus increasing the electric field. We can increase the dielectric constant to improve the activity of the reaction and increase the temperature of the reactor [86,87].

In DBD plasma, different dielectric materials can often be used as dielectric barriers. Quartz and alumina are two representative dielectric materials affecting the discharge behavior of DBD plasma and the dissociation of $CH_4$ [86,88]. Compare with quartz, alumina has a higher dielectric constant and porous morphology. At the same time, the dielectric material also affects the discharge behavior with respect to the applied voltage and discharge current. Due to the high discharge power of the reactor in alumina, the high electric field generated by its partial porous structure helps to activate the reactant species in the pores of the alumina material.

### 4.3. Discharge Volume

The $V_D$ of the DBD plasma reactor is composed of $D_{gap}$ and DL [89]. $D_{gap}$ can be expressed by using the difference between the inner radius of the dielectric and the outer radius of the high-voltage electrode. Duan et al. investigated the effect of $D_{gap}$ on $CO_2$ conversion in DRM reaction and found that a larger $D_{gap}$ could increase the $CO_2$ conversion due to the increased residence time of $CO_2$, leading to a prolonged contact with other reactive species [90]. In another study, 3 mm $D_{gap}$ realized the highest conversion of $CO_2$ and methane when $D_{gap}$ varied from 1–5 mm, suggesting the necessity of optimal selection of an appropriate $D_{gap}$ value [87].

DL is another geometric parameter in DBD plasma. The length of the electrode can affect the value of DL and corresponding VD. Relevant studies suggested that increasing DL can increase the conversion rate, which is due to the higher contact time of the reactive species and gas molecules at a constant flow rate [87].

## 5. Effects of Process Parameter

Process parameters exert crucial influences on realizing the industrialization of technology. Many studies have described the process parameters, such as input power (Pin), feed flow rate, specific input energy (SIE), and feed ratio [69,91,92]. These parameters have a certain impact on gas processing and system efficiency. Therefore, by adjusting these process parameters, it can help to allocate targeted products reasonably.

### 5.1. Input Power

Input power is an important process parameter of catalytic DRM reaction by DBD plasma. It is related to the energy efficiency (EE) of the gas treatment and processing. Generally speaking, for a DBD plasma reactor, as the input power increases, the $CH_4$ and $CO_2$ conversion rate increases. This is because the increased input power can increase the electron density, accelerate the collision of the reaction gas molecules with high-energy electrons, and promote the activation of the reactants. These excited, dissociated, and ionized reactant molecules triggers the dry reforming reaction of methane [89].

The discharge power can be adjusted by adjusting the voltage (V) and frequency (f). As the voltage and frequency increase, the current pulse increases, thereby increasing the processing of the reactant gas. For example, Liu et al. found that the selectivity of $H_2$ and CO in DBD increases with the increase of discharge power [69]. On the other hand, the frequency can change the reactivity within a certain range; once this range is exceeded, the

conversion rate will no longer be affected by the frequency [87,93]. A similar phenomenon can also be seen at a voltage proportional to the plasma discharge power.

### 5.2. The Feed Flow Rate

The feed flow rate is another vital process parameter in the DBD plasma reactor, considering the impact on the conversion of the $CO_2$ and $CH_4$. Many studies have shown that various parameters are related to the total flow rate of the feed [94]. For example, Eliasson et al. studied the feed flow range of 200 to 900 mL min$^{-1}$ and found that $CO_2$ and $CH_4$ conversions dropped from 54% and 60% to 18% and 23%, respectively [8]. Rico and colleagues found that with the increase of flow rate, $CH_4$ and $CO_2$ conversions decreased. However, the selectivity of $H_2$ and CO is not affected despite the highly affected yield.

The flow rate of feed is also linked to the mass transfer limitation and processing capacity of the DBD plasma reactor, and is inversely proportional to the reactants' residence time in the discharge zone [87]. The processing capacity of the reactor can be improved by adding a potential catalyst in DBD plasma. Mass transfer limitations can be overcome by high flow rates. At higher flow rates, the external mass transfer limitation will be minimized, but high flow rates will also cause some adverse effects [93,95].

### 5.3. Specific Input Energy

Specific input energy (SIE) refers to the energy required for processing raw gas into products or the energy required per liter of raw gases. SIE can be tuned by changing the input power and the total gas feed flow rate. Pinhao et al. studied DBD plasma methane dry reforming and found that the flow rate of feed and SIE are inversely proportional to each other. Therefore, $CH_4$ and $CO_2$ conversions are directly related to SIE [91]. With feed flow rate unchanged, the SIE increases with the increase of the input power, which in turn leads to an increase in catalytic performances, while the system efficiency of the DBD plasma reactor decreases [59,96].

### 5.4. The Feed Ratio

The feed ratio affects the selectivity $H_2$. Figure 8 presented the $CO_2/CH_4$ molar ratio effect on the DRM performance in a DBD plasma reactor without catalyst packing. In Figure 8, as the $CO_2/CH_4$ ratio increased, the $CH_4$ conversion and CO selectivity increased almost linearly. When the $CO_2/CH_4$ ratio increased from 1:9 to 1:3, the $CO_2$ conversion dropped from 41.5% to 21.3%; however, when further increasing the ratio to 9:1, the $CO_2$ conversion slightly increased as shown in Figure 8a. In addition, by increasing the $CO_2/CH_4$ ratio further to 9:1, the $H_2$ selectivity was close to 100%, and the CO selectivity increased by 8.8 times, while the $C_2H_6$ selectivity decreased by 53% (Figure 8b) [27] (Table 1). These studies paved the way to optimize the conversion and selectively by adjusting the feed ratio.

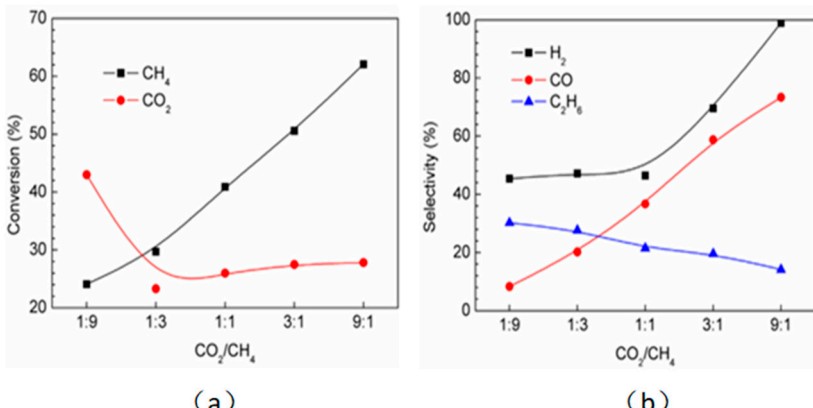

**Figure 8.** The $CO_2/CH_4$ molar ratio effect on the catalytic performances of plasma dry reforming of methane (DRM) reaction: (**a**) Conversion; (**b**) selectivity. (Total feed flow 25 mL min$^{-1}$, discharge power 15 W). Reproduced with permission from [27], Copyright 2015, Elsevier.

## 6. Conclusions and Outlook

This article reviews the recent development of DRM catalyzed in DBD plasma reactors on Ni-based catalysts, including an overview of non-thermal plasma, the mechanism of DRM with different plasmas, the synergy of plasma and Ni-based catalyst, reaction design, and process parameter optimization. The synergistic effect of DBD plasma and different Ni catalysts are classified and discussed in two categories: Pure Ni with various supports and Ni-based catalysts with doping. The synergistic effects of DBD plasma and pure Ni catalyst include size reduction of Ni grains, promoted discharge heating, higher efficiency of the DBD plasma reactor, and enhanced reactant adsorption. By doping with other metals and oxides, a higher active phase dispersion and stronger metal-support interaction are realized; the alkalinity of the catalyst is increased to promote the activation of $CO_2$ and $CH_4$; due to the higher dielectric constant, the dielectric performance of the plasma is better. Although great progress has been made in this area, the technology still faces many challenges, and there are still areas for further improvement:

(1) For DBD plasma-catalytic DRM, one challenge is power dissipation and carbon deposition. The trend of coke formation can be observed through the carbon balance and $H_2/CO$ ratio [97]. The $H_2/CO$ ratio close to unit 1 is considered to be an ideal method for processing syngas, and a lower carbon balance suggests a higher coke deposition. Carbon deposition in DRM is not only due to side reactions, but also due to the feed ratio of $CH_4/CO_2$ and other parameters [21]. Although catalysts have been used to avoid the formation of excess carbon and increase the reaction rate, the problem of carbon deposition has not been completely eliminated. Therefore, we continue to explore solutions to achieve better selectivity and stability.

(2) The synergy between the plasma and catalyst has been confirmed in recent years, but the interaction between the two has not been fully explored. Advanced methods are needed to study the mechanisms in depth to realize the rational design of the catalyst-plasma system [98].

(3) Efficient dry reforming of methane provides excellent energy efficiency, which is worthy of further consideration and research. In order to industrialize plasma catalysis, it is necessary to have a deeper understanding of the relationship between the catalytic performance and energy efficiency vs. reactor configuration and process parameter.

**Author Contributions:** X.G.: Conceptualization, data curation, investigation, writing—original draft, writing—review and editing; Z.L.: Conceptualization, data curation, investigation, writing—original draft, writing—review and editing; T.L.: Writing—original draft; L.H.: Writing—original draft; J.Z.: Writing—review and editing; S.A.: Data curation, writing—review and editing; N.D.: Data curation, writing—review and editing; A.J.: Project administration, supervision, validation; S.K.: Funding

acquisition, resources, project administration, supervision, validation. All authors have read and agreed to the published version of the manuscript.

**Funding:** This research was funded by Ministry of Education in Singapore (MOE) Tier 2 grant (WBS: R279-000-544-112), Singapore Agency for Science, Technology and Research (A*STAR) AME IRG grant (No. A1783c0016), National Environment Agency (NEA) in Singapore (WTE-CRP 1501-103) and Youth Innovation Talents Project of Guangdong Universities (natural science) in China (2019KQNCX098).

**Data Availability Statement:** All data included in this study are available upon the permission from the publishers.

**Conflicts of Interest:** The authors declare no conflict of interest.

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
