# Peer review of "Recent Developments in Dielectric Barrier Discharge Plasma-Assisted Catalytic Dry Reforming of Methane over Ni-Based Catalysts"

_catalysts, doi:10.3390/catal11040455_

Round 1

Reviewer 1 Report

The review b Gao et al. is a very nice contribution into the field of non-thermal plasma assisted catalytic processes. The authors focus particularly on the methane reforming reaction by CO2 using Ni catalyst. The paper is very well written, contains 100 references. The process is very actively developing so I think the paper is timely.

As a very minor change I would suggest the removal of "Bob Dudly of BP" statement. Alternatively, a source of a quote should be provided.

Abbrev. DRM should be introduced in the intro, not only in the abstract.

Author Response

To Reviewer #1:

Comment: The review by Gao et al. is a very nice contribution into the field of non-thermal plasma assisted catalytic processes. The authors focus particularly on the methane reforming reaction by CO2 using Ni catalyst. The paper is very well written, contains 100 references. The process is very actively developing so I think the paper is timely.

Response: Thank you for your positive comments and valuable suggestions to improve our manuscript. The responses are listed as below.

Q1: As a very minor change I would suggest the removal of "Bob Dudly of BP" statement. Alternatively, a source of a quote should be provided.

Response: Thank you for your valuable suggestions to improve our manuscript. The statement has been removed accordingly.

Q1: Abbrev. DRM should be introduced in the intro, not only in the abstract.

Response: Thank you for your good comments to improve our manuscript. The DRM is introduced in the introduction part.

Reviewer 2 Report

see attached file

Author Response

To Reviewer #2:

Comment: The manuscript presents a review on dry reforming of methane (DRM) by plasma of dielectric barrier discharge (DBD) assisted by various Ni-based catalysts. The review thoroughly summarizes the main and the most recent scientific papers, results and findings in the field of DRM. The presentation of the results is not always ideal, sometimes confusing, inconsistent and often hard to follow. It is quite surprising as the authors are experienced researchers who published many papers relevant to the topic in various journals [3-7. 11-16, 24-29]. I recommend the manuscript for a detailed revision as I believe the presentation of the data and the whole concept of the review can still be significantly improved.

General comment 1: Information on physics of the discharges is difficult to understand even to an expert in  the field not to mention a generic reader. It must be definitely improved (see comments below).

General comment 2: There is a huge number of papers and results mentioned and discussed in the manuscript. With such a huge number it is very important to make it clear for a reader what is a train of thought authors follow and the conclusion which comes out of it. I found it often difficult to follow the results and ideas in certain sections and paragraphs. While referring to individual studies authors present a set of many parameters (type of power supply, temperature, type of catalyst, its loading, porosity, surface area, working gas, efficiency, etc.), however not always the same set of parameters for each study. So it is quite difficult to compare individual results and understand their importance. Some works are mentioned only briefly (in one sentence), while others are refereed in huge details even exceeding a half of a page (e.g. [52]). Studies on DRM are usually mentioned one by one, not always integrated, compared and confronted. There is a Table 1 at the end of the paper that is absolutely not referred in the main text. Not just once. The table should list also lits other important parameters, such at temperature, specific input energy or even energy efficiency, and also yield of H2 and CO (their selectivity) and eventually also input concentrations of CH4 and CO2.

Response: Thank you for your very detailed comments and valuable suggestions to improve our manuscript. The responses are listed as below.

Q1: Title should be updated and include ‘dielectric barrier discharge’ as the review is mainly dedicated to its effects. It is also clearly mentioned by the authors in Abstract and Conclusions: In this review, the recent developments of catalytic DRM in DBD plasma … are summarized.

Response: Thank you for your comments as well as valuable suggestions to improve our manuscript. The new title have been updated as below:

Recent developments in dielectric barrier discharge plasma-assisted catalytic dry reforming of methane over Ni-based catalysts.

Q2: Methane reforming by CO2 (DRM). It is not appropriate way define an abbreviation. I would recommend to use ‘dry reforming of methane’ (DRM) always.

Response: Thank you for your valuable suggestions to improve our manuscript. The statements have been updated in abstract and introduction.

Q3: dry methane reforming (DRM) … please update the order of words.

Response: Thank you for your good comments to improve our manuscript. The keyword has been revised.

Q4: the interaction between the catalyst and active species increases the lifetime and collision probability of the reactive species, and enhances the surface modification and electric field [8] ... How a collision can increase lifetime of reactive species and enhances electric field?

Response: Thank you for your enlightened comments to improve our manuscript. Please kindly allow us to clarify that according to the original sentence “the interaction between the catalyst and active species increases the lifetime and collision probability of the reactive species, and enhances the surface modification and electric field”, it is the “interaction” instead of “collision” that enhances both lifetime and electric field. To make it clearer, a revision of this part is given as below and highlighted at line 75 in page 2:

The electric field of plasma is enhanced due to charge accumulation and polarization effect caused by the roughness of catalysts. Also, a higher interaction between the catalyst and active species causes a higher conductivity which improves the magnitude of the electric field. Meanwhile, the catalyst facilitates the adsorption and prolongs the contact time of reactants, leading to a higher activation and conversion. On the other hand, the highly reactive species generated by plasma cause the structure change and surface faceting of the catalyst, promoting the charge deposition and hotspot formation, which activates the metals and reactants. The combination of plasma and catalysts can prolong the lifetime of active species and lower the activation barrier.

Q5: Among the catalysts for plasma-catalyzed DRM, noble metals show good catalytic performances ....Please list the most typical precious metals that have been used for plasma assisted DRM. Please, briefly characterize the best results achieved.

Response: Thank you for your good suggestion. The most typical precious metals that have been used for plasma assisted DRM are added in page 2. The statements are also shown as below:

As seen from previous studies, precious metals show strong resistance to coke formation and exhibit high catalytic activity in thermal and plasma DRM. The precious metals commonly used in plasma DRM include Pt、Pd、Rh and Ru [8]. For example, Pt nanoparticles were impregnated in the UiO-67 MOF structure, accelerating the reactant dissociation and enhancing H2 yield. Moreover, due to the dehydrogenation of hydrocarbons on Pt nanoparticles, the selectivity towards light hydrocarbons was reduced by 30%. Assisted by plasma, the surface reactions were intensified, increasing the energy efficiency by 11%. Furthermore, excellent stability was shown by the constant conversions during four cycles of plasma on-off [23].

Q6: Among the non-precious metals, Ni-based catalysts are currently considered one of the most promising catalysts in DRM reactions due to the high catalytic activity and low cost [25-29]. Please list other non-precious metals that are being used besides Ni. Also please state, why Ni is considered to be the most promising out of them. You can even present a short overview of the results achieved with other than Ni- catalyst so the reader understand the potential and advantages of using Ni, when compared with other non-precious metals and also to understand the importance of this review.

Response: Thank you for your comments and valuable suggestions to improve our manuscript. The responses are listed as below and also highlighted in page 2-3:

Relatively speaking, transition metals such as Co、Mn、Fe、Cu, Ni are also used for plasmonic catalytic DRM reactions. Zeng et al. compared γ-Al2O3 supported metal catalysts M/γ-Al2O3 (M = Ni, Co, Cu and Mn) with the help of plasma. They found out that the combination of plasma with Ni/γ-Al2O3 and Mn/γ-Al2O3 catalysts significantly increased CH4 conversion [26-27]. Other studies have shown that among various transition metals (Ni、Co and Fe) used to catalyze DRM, Ni based catalysts presented good activities and economic feasibility [28-31], currently considered one of the most promising catalysts in DRM reactions due to the high catalytic activity and low cost [32-36].

Q7: For example, Tu, et al. Studied. I would not cite these works already here. Discussion related toresults achieved using various Ni-based catalysts should rather be a part of the following chapters. In Introduction I would rather like to see a brief overview on various precious and non-precious metals for plasma assisted DPM including slowly converting to Ni-based catalysts.

Response: Thank you for your comments and valuable suggestions to improve our manuscript. The two examples for ref 30-31 have been removed.

Q8: Therefore, this article summarizes the methods of using different types of NTP to catalyze DRM, discusses Although the Abstract and Conclusion state the review is about DBD assisted catalysis for DRM,here the authors claim different. I would recommend the manuscript focus on DBD only as there is very little information of DRM by other discharges.

Response: Thank you for your good comments to improve our manuscript. The revised sentence in the last paragraph of introduction is shown as below and highlighted in page 3:

This article focuses on the application of DBD and Ni catalysts to catalyze DRM.

Q9: Please consider to reduce to this part of the manuscript. I think the review should strictly focus on DBD+Ni-catalysts vs. DRM. Characterization of basic concept of NTP can be shortened and information on various types of discharges can be completely removed as they are barely discussed with respect to DRM (see below). Further, the information on electric discharges should have been written by someone who understands the physics of discharges and the used terminology. To me it seems it was written by someone who is not an expert as the text contains many strange expressions and even non-sense. Bellow are few examples (bold are words/expression is a question).

Response: Thank you for your valuable comments to improve our manuscript. Accordingly, the characteristics of NTP are shortened and other types of discharges have been removed.

Q10: NTP generation method is to generate electricity by applying a high potential difference ?

Response: Thank you for your good comments and valuable suggestions to improve our manuscript. The revised content is listed as below and highlighted in page 3.

The widely accepted and facile NTP generation method is by means of electricity. Capacitively coupled plasma is a typical example, where a large potential difference is exerted between two electrodes. Due to the electrical discharge generated by the electric field, the gas between the electrodes is transformed into plasma.

Q11: The intensity of the gas electric field is high enough to accelerate the electrons?

Response: Thank you for your valuable suggestions to improve our manuscript. The responses are listed as below and highlighted in page 3-4.

Considering the thousands of volts voltage and short distance between electrodes, the intensity of electric field in gas is high enough to accelerate the electrons from one electrode to another, where a very high energy can be transformed [2].

Q12: Influenced by the electric field shielding, a discharge is caused. what do you mean by field shielding?

Response: Thank you for your valuable comments to improve our manuscript. The statement has been removed.

Q13: GA is a warm plasma source for synthesizing ... what is ‚warm? There is no such a term as‚warm plasma.

Response: Thank you for your valuable suggestions to improve our manuscript. The statement has been removed.

Q14: Microwave discharge is warm plasma and a non-polar discharge discharge, generated when exposed to electromagnetic waves. Warm plasma? Non-polar discharge? Discharge is generated when exposed to EM waves? A discharge, not a gas exposed?

Response: Thank you for your comments and valuable suggestions to improve our manuscript. The statement has been removed.

Q15: Electrons in light-emitting power generation usually collide with and excite the neutral atoms or molecules.

Response: Thank you for your valuable comments and valuable suggestions to improve our manuscript. The statement has been removed.

Q16: Corona discharge, whose name comes from the streamer crown. A corona discharge has various modes depending on the polarity and the applied voltage. Its name derives rather from glow corona mode, rather than a streamer corona mode. I never heard of a ‘streamer crown’.

Response: Thank you for your valuable comments and valuable suggestions to improve our manuscript. The statement has been removed.

Q17: Corona discharge, ... is a partial self-sustained discharge with a gas medium near an electrode with a small curvature radius . It sound like a gas is only near the electrode with a small curvature.

Response: Thank you for your valuable comments and valuable suggestions to improve our manuscript. The statement has been removed.

Q18: uneven electric field ... ‘non-homogenous field’ is a term usually used.

Response: Thank you for your valuable comments and valuable suggestions to improve our manuscript. The statement has been removed.

Q19: Its formation usually starts with high DC voltage applied to both ends of the electrode.

Response: Thank you for your valuable comments and valuable suggestions to improve our manuscript. The statement has been removed.

Q20: The discharge gas of CD fits the applications in a large scale. No idea what this sentence means.

Response: Thank you for your valuable comments and valuable suggestions to improve our manuscript. The statement has been removed.

Q21: Below, types of NTPs, working principles and operating conditions are introduced with the highlights of their advantages and disadvantages and illustrations of their applications to DRMs. Authors claim that in chapters 2.3.1 – 2.3.6 present 1) working principles, 2) operating conditions, 3) advantages and disadvantages and 4) illustrations of their applications to DRMs. However this is not true. It is different from chapter to chapter. The working principles are often not well explained or not explained at all. For example in case of GA 1) and 4) are completely missing. In case of MW discharge and corona discharge 4) i.e. applications toward DRM are also not mentioned. In case of RF discharge 3) is not commented. As already stated above, many information and expressions in these chapters are very difficult to understand.  

Response: Thank you for your valuable comments and valuable suggestions to improve our manuscript. Chapter 2.3.1 – 2.3.5 have been removed.

Q22: In recent years, microwave discharge plasma has been widely used in material preparation and fuel reforming to produce hydrogen ... reference is needed.

Response: Thank you for your valuable comments and valuable suggestions to improve our manuscript. The statement has been removed.

Q23: The MW discharge electric field is usually low, and electrons with energy of about 1 eV are generated. reference for the whole paragraph is needed.

Response: Thank you for your good comments and valuable suggestions to improve our manuscript. The statement has been removed.

Q24: Recently, the use of MW plasma for CO2 reforming has received increasing attention....... reference is needed.

Response: Thank you for your good comments and valuable suggestions to improve our manuscript. The statement has been removed.

Q25: Glow discharge plasma is mostly used for solid surface treatment, especially for catalysts....... reference is needed.

Response: Thank you for your good comments and valuable suggestions to improve our manuscript. The statement has been removed.

Q26: As mentioned above I think the manuscript should solely focus on the DBD+Ni-catalysts vs. DMR. The principles of DBD must be clearly explained. Its common geometry and operation conditions (used power supply, pressure, gas mixtures, etc. ). Also the expressions must be mastered and improved.

Response: Thank you for your academic comments and valuable suggestions to improve our manuscript. The responses are listed as below and highlighted in page 4.

Besides, Dielectric barrier discharge (DBD) is a unique plasma device due to the dielectric material layer between the electrode and gas, which can pass in one direction through a limited amount of current, named displacement current. When electricity is applied, plasma can be generated via the charge accumulation on the dielectric material with the short lifetime streamer, which will stop the discharge at one point, thus driving the discharge to happen at another point on the surface [41].

Q27: DBD is a technology to produce electrons with a high energy to activate greenhouse gases which are usually chemically stable.

Response: Thank you for your good comments and valuable suggestions to improve our manuscript. The responses are listed as below and highlighted in page 4.

Moreover, DBD is a technology to generate stable plasma in a large scale at a low temperature and to produce electrons with a high energy to activate and dissociate greenhouse gas molecules such as CH4 and CO2 which are usually chemically stable due to the high bonding energies.

Q28: Dielectric barrier discharge has the characteristics of both corona discharge high-voltage operation and large-space uniform discharge of glow discharge.

Response: Thank you for your valuable comments and suggestions to improve our manuscript. The responses are listed as below and highlighted in page 4.

On the other hand, Dielectric barrier discharge can generate stable and uniform atmospheric pressure plasma under atmospheric pressure or higher. After optimization, this type of discharge has a broad development prospect in the industry.

Q29: Typical DBD reactor is composed of two electrodes, asymmetrically located on top and bottom sides of the dielectric barrier material, ... vs. 575: The shape of the DBD plasma reactor is cylindrical ?

Response: Thank you for your valuable comments and suggestions to improve our manuscript. The statements in 2.3 have been moved to 4.1 as below:

Commonly used reactor configurations in DBD plasma include planar and cylindrical types. Typical planar DBDs reactor is composed of two electrodes (as shown in Figure 5), asymmetrically located on top and bottom sides of the dielectric barrier material, which limits the current and inhibits the sparks formation. Electric discharge is generated when an AC potential is exerted [24,26,65].

Figure 5 Two frosted glass plates are used to mix parallel plate DBD reactors. Reproduced with permission: Copyright 2020, Springer.[65]

Q30: DBD plasma is a means to effectively activate molecules. This is a common feature of all discharges not just DBD.

Response: Thank you for your valuable comments and suggestions to improve our manuscript. The responses are listed as below. Plasma can effectively activate molecules is one of the common features of all discharges, which is worthy of recognition. We emphasize that DBD plasma can activate molecules effectively in order to further illustrate the advantages of DBD plasma, especially for DRM reaction.

Q31: Reduce the requirements for experimental equipment?

Response: Thank you for your valuable comments and suggestions to improve our manuscript. The statement has been removed.

Q32: a gentle reactor design?

Response: Thank you for your valuable comments and suggestions to improve our manuscript. The statement has been removed.

Q33: the distribution of CH4 discharge products is wide?

Response: Thank you for your valuable comments to improve our manuscript. The statement has been removed.

Q34: Although DBD plasma has certain advantages over DRM, low energy utilization efficiency caused by the reducing electric field limits the application.

Response: Thank you for your valuable comments to improve our manuscript. The responses are listed as below and highlighted in page 4.

Despite the advantages of DBD plasma over DRM, low energy utilization efficiency limits the application [10,21,26].

Q35: A reference is missing.

Response: Thank you for your valuable comments to improve our manuscript. The statement has been removed.

Q36Discussion on plasma-catalyst synergistic effect is often a problem. Especially when plasma catalytis reactor is operated with a high input energy (as it happens also in DMR systems) that leads to the heating of the catalyst (C) that can become reactive even without plasma (P). Many author claim synergy when the efficiency of P-C system is better than the sum of the individual systems (P and C) without considered the fact that in P-C system the catalyst temperature is not identical to that in C system. Therefore the synergy must be carefully considered not to overemphasize the achieved results in P-C system and claim synergy although there is none. Please be careful while talking about synergy and always present it with respect to the actual temperature of the catalyst/system. Btw, in my opinion in this chapter there is no actual discussion on synergy itself. The word ‘synergy’ is mentioned just once and the whole chapter.

Response: Thank you for your valuable comments and suggestions to improve our manuscript. The additional discussion of the synergy is listed as below and highlighted in page 5.

In a word, the synergy between catalysts and DBD plasma can be summarized based on Figure 1 as below: first, the carbon deposition is alleviated; second, the conversions, selectivity and stability are enhanced; third, catalyst reusability is improved; fourth, energy efficiency is increased [8].

     Figure 1 Synergy of the catalytic DBD plasma system. Reproduced with permission: Copyright 2019, Elsevier.[8]

Q37Plasma physically and chemically affects the surface area and interfacial chemistry of the catalyst which in turn enhances DRM activity and improves product distribution. It is not the only the plasma that affects the catalyst. The catalyst itself (based on its shape, composition etc.) may affect the plasma properties (discharge type), its distribution in volume and on surface and thus the production of reactive species. Please. address also this point.

Response: Thank you for your valuable comments and suggestions to improve our manuscript. The responses are listed as below and highlighted in page 4.

Charge accumulation and polarization effect caused by the various shape and surface properties of catalysts enhance the electric field of plasma [8].

Q38: The roughness of the catalyst has a certain impact on it ... Please, be more specific. How is roughness defined? In case of catalysts properties like pore size, porosity, specific surface area, etc. usually characterize surface and its morphology. Please refer to them.

Response: Thank you for your valuable comments and suggestions to improve our manuscript. The responses are listed as below and highlighted in page 5.

The roughness of the catalyst has a certain impact on it, due to the enhanced polarization effect and accumulation of charge in a more dispersed and smaller particle, high surface area and high density of metal sites [48].

Q39: the sintering of nickel catalyst under high temperature reaction is one of its main disadvantages. How much is ‚high‘?

Response: Thank you for your valuable comments and suggestions to improve our manuscript. The temperature is generally above 500 ℃.

Q40: Figure 2: Spent catalyst. How much spent? How long it has been used (GHSV)?

Response: Thank you for your good comments as well as valuable suggestions to improve our manuscript. The reaction lasts for 5 h.

Q41: The plasma-decomposed Ni/ZrO2 possesses. What is plasma-decomposed catalyst?

Response: Thank you for your valuable comments to improve our manuscript. The responses are listed as below and highlighted in page 7.

The nickel precursor is decomposed under the DBD plasma to prepare a nickel/zirconia catalyst, possessing a high specific surface area and provides more surface Ni active centers for the reaction.

Q42: Terms Ni/ZrO2-C and Ni/ZrO2-P are not explained.

Response: Thank you for your good comments and valuable suggestions to improve our manuscript. The details are listed below and highlighted in page 7:

In ZrO2-P (precursor decomposed with plasma treatment in Ar atmosphere), the amount of t-ZrO2 is twice that of Ni/ZrO2-C (precursor calcined in air without plasma treatment), indicating the beneficial effect of NTP treatment on the formation of t-ZrO2 [56].

Q43: Is seems the whole chapter 3.1.3. is based on the results of just one paper [52]. I found it a little strange to refer to a single works in details on more than half on a page, while other important works are mentioned only briefly and the related text suffers from lack of information on exact experimental condition and achieved results, Please consider this fact here and elsewhere.

Response: Thank you for your valuable suggestions to improve our manuscript. The statements are also shown as below and highlighted in page 7:

ZrO2 has been studied as the packing materials in DBD plasma reactor to catalyze the DRM reaction. However, with ZrO2 alone, the fraction of void decreased, reducing the current intensity and plasma generation. Moreover, surface discharges became dominant instead of filamentary microdischarges, negatively affecting the activation of reactant molecules [23].

In contrast, Ni/ZrO2 exhibits an enhanced reactivity in DBD DRM reaction. The nickel precursor is decomposed under the DBD plasma to prepare a nickel/zirconia catalyst, possessing a high specific surface area and provides more surface Ni active centers for the reaction. During the early work, the application of Ni/ZrO2 catalysts in dry reforming has been studied, and it is found that carbon deposition is significantly reduced, especially under low nickel loading, if the nickel microcrystals used are very small [67]. ZrO2 contains two crystal phases, namely t-ZrO2 and m-ZrO2. In ZrO2-P (precursor decomposed with plasma treatment in Ar atmosphere), the amount of t-ZrO2 is twice that of Ni/ZrO2-C (precursor calcined in air without plasma treatment), indicating the beneficial effect of NTP treatment on the formation of t-ZrO2 [56]. Compared with Ni/ZrO2-C, the agglomeration of nickel particles on the Ni/ZrO2-P was much reduced, and the exposed surface lattice fringes of the Ni particles were clearly visible. In addition, more oxygen vacancies were generated on Ni/ZrO2-P, which provided stronger alkalinity and promoted CO2 adsorption and activation, enhancing the activity of Ni/ZrO2 in DRM. Meanwhile, oxygen vacancies additionally drives the CO2 reduction, resulting in enhanced activation of CO2 and Ni/ZrO2 DRM activity [56,68]. The methane and CO2 conversions using plasma and catalyst were 53.57% and 60.81%, higher than 42.30% and 52.88% with plasma alone, indicating that the combined treatment of DBD plasma and Ni/ZrO2 can provide stronger adsorption sites and improve the adsorption capacity of CO2, further increasing the activity of Ni/ZrO2 in DRM. Finally, by comparing the carbon deposition amount of the two catalysts after the DRM reaction, Ni/ZrO2-P possessed lower coke formation (The carbon deposition amount of Ni/ZrO2-P and Ni/ZrO2-C is 33 wt% and 64 wt% respectively) [56].

Q44: In summary, studies have confirmed that DBD plasma ....there is just one study mentioned in the chapter 3.1.3 [52], not several studies.

Response: Thank you for your valuable suggestions to improve our manuscript. We have made revisions according to your suggestions. The details are listed below and highlighted in page 7.

In summary, studies by Vakili and Hu et al. [23,56] have confirmed that DBD plasma decomposition benefits the formation of highly active Ni-based DRM catalysts.

Q45: Moreover, metals such as Co, K, Fe, Mg, Mn, and Ce have been used as modifiers recently. Transition metals (Co, Mn, Fe), alkali metals (Mg, K) and rare earth metals (Ce, La) are mentioned here. It would be nice to mention these metals also at the beginning of the respective chapter 3.2.x, so the reader knows what metal will be discussed in the chapter. By the way, Fe is mentioned here, but no reference in the manuscript can be found. Can you comment?

Response: Thank you for your good comments as well as valuable suggestions to improve our manuscript. The statements are also shown as below and highlighted in each section (Fe has been removed from the statement):

Transition metals have drawn interest as a doping agent to promote the performance of Ni catalysts for DBD-catalyzed DRM reaction, such as Mn [20] and Co [70].

In addition to transition metals, the addition of promoters such as alkali and alkaline earth metals (Mg and K [61,63])

Similarly, rare earth metals such as La and Ce can act as a doping agent to enhance the catalytic performance of Ni-based catalysts in DBD reactor for DRM [73-74].

Q46: Moreover, without the promoter K, the yield of H2 was only 9% . This is a very isolated case when H2 yield is mentioned in the manuscript. With respect to DRM the products yield and their selectivity are very important parameters. I think they should be mentioned in case of other works too and compared.

Response: Thank you for your valuable suggestions to improve our manuscript. We have made revisions according to your suggestions. The details are listed below and highlighted in page 9.

Compared with plasma alone, integrating plasma with Ni-K/Al2O3 catalyst greatly increased CH4 and CO2 conversion to 31.6% and 22.8% respectively. Moreover, H2 selectivity was enhanced to 43.3% [63].

Q47: When the reactants collided with high-energy substances, adhesion,  excitation, dissociation and ionization process produced intermediates and final products ??? What reactants? What substances? What processes and what kind of intermediate and final products? Please understand that a sentence like this one brings more questions than gives answers to a reader.

Response: Thank you for your valuable suggestions to improve our manuscript. The statements are shown as below and highlighted in page 12:

When the reactant gas molecules collided with high-energy electrons, the electrons in the molecules were excited, dissociated and ionized to produce active species such as O, H, methyl radicals, and finally CO and H2.

Q48: DBD plasma reactor design. Please explain what is the main objective of this chapter. It was not introduced in the Introduction part among the objectives.

Response: Thank you for your valuable suggestions to improve our manuscript. The statements are shown as below and highlighted in page 12:

To achieve a high catalytic performance and energy efficiency, reactor designs including the configuration of the reactor, medium materials and discharge volume play an essential role. In the following part, DBD plasma reactor designs will be illustrated in detail.

Q49: Configuration of DBD plasma reactor. What kind of reactor is this? In chapter 2.3.6 you mentioned Typical DBD reactor is composed of two electrodes asymmetrically located‘, while here and also in the caption of Fig. 5 you state ‘The typical geometry of a DBD plasma reactor system’ which is however not planar but a cylindrical geometry. It is confusing and inconsistent. In addition Fig. 5 presents various plasma reactors but the most common DBD reactor for plasma assisted catalysis is a reactor packed with the pellets (packed-bed reactor), which is not depicted.

Response: Thank you for your good comments a well as valuable suggestions to improve our manuscript. Additional statements are shown as below and highlighted in page 12 and page 14:

Commonly used reactor configurations in DBD plasma include planar and cylindrical types. Typical planar DBDs reactor is composed of two electrodes (as shown in Figure 5), asymmetrically located on top and bottom sides of the dielectric barrier material, which limits the current and inhibits the sparks formation. Electric discharge is generated when an AC potential is exerted [24,26,65].

Figure 5 Two frosted glass plates are used to mix parallel plate DBD reactors. Reproduced with permission: Copyright 2020, Springer.[65]

As the most common DBD reactor for plasma-assisted catalysis, packed-bed reactors with different designs are shown in Figure 7 [8]. Pre-packing mode is mostly applied in those temperature-controlled reaction. The catalysts needs to be activated first, followed by passing the reactants to the discharge zone, which may not be practical in DBD-DRM system. In contrast, in-situ packing reactor takes advantages of the catalyst and plasma simultaneously. With a relatively stable temperature, the excited electrons promote the dissociation of reactant molecules. Few researchers focus on the post packing and fully packed DBD reactor in DRM reaction since they might be not effective as the in-situ packing mode.

Figure 7 Different reactor configurations based on the packing material. Reproduced with permission: Copyright 2019, Elsevier.[8]

Q50: A reference is missing

Response: Thank you for your good comments to improve our manuscript. Reference has been added accordingly in page 13.

Q51: Quartz and alumina are two representative dielectric materials. Is it the material of the tube around which the electrode is wrapped or is it a material that covers a central electrode?

Response: Thank you for your good comments a well as valuable suggestions to improve our manuscript. The statements are shown as below:

The central electrode materials are fitted centrally along the inside of the dielectric materials and the outer electrode materials are rolled around the outside of tubular dielectric materials.

Q52: Alumina has a ... higher electric field capacity ???

Response: Thank you for your valuable suggestions to improve our manuscript. The statements are shown as below and highlighted in page 14:

Compare with quartz, alumina has a higher dielectric constant and porous morphology.

Q53: The VD of the DBD plasma reactor. VD Ba careful with subscripts, here and elsewhere.

Response: Thank you for your valuable suggestions to improve our manuscript. The statements are also shown as below and highlighted in page 14:

The VD of the DBD plasma reactor is composed of Dgap and DL [94].

Q54: CO2 conversion in DRM reaction and found that a smaller Dgap could increase the CO2 conversion due to the intimate contact between CO2 and reactive species. Explanation needed.

Response: Thank you for your valuable suggestions to improve our manuscript. The statements are shown as below and highlighted in page 14:

Duan et al. investigated the effect of Dgap on CO2 conversion in DRM reaction and found that a larger Dgap could increase the CO2 conversion due to the increased residence time of CO2, leading to a prolonged contact with other reactive species [95].

Q55: I expected the chapter to summarize the current knowledge on DRM by DBD-catalytic systems based on the results presented in the previous chapters and as a next step to discuss their optimization. I do not think the chapter is about real optimization, but rather comment on the effect of selected operating parameters (input power, SIE, gas flow etc) on DRM.

Response: Thank you for your good comments and valuable suggestions to improve our manuscript. The title of this chapter has been changed accordingly.

Q56: What is active material? Is the active material the same thing as active substance in the following sentence? What are they? What are gas particles? Can you be more specific in your expressions? Again here, it is really difficult to guess what do you mean..

Response: Thank you for your positive comments as well as valuable suggestions to improve our manuscript. The statements are shown as below and highlighted in page 15:

This is because the increased input power can increase the electron density, accelerate the collision of the reaction gas molecules with high-energy electrons, and promote the activation of the reactants. These excited, dissociated and ionized reactant molecules triggers the dry reforming reaction of methane [94].

Q57: It is a pitty that such important parameter as H2 selectivity which is critical for the industrial process of DMR is first time mentioned in the manuscript just a half page before Conclusions. I think more space should be dedicated to H2 selectivity. At least in Table 1, but also in the main text if possible.

Q58: As mentioned above I would recommend to expand the table (maybe to turn it by 90 degrees, if the MDPI format allows it) and includes parameters like temperature, SIE, energy efficiency, H2 and CO2 yield and selectivity. Remarks can be minimized or included in the main text. Also the table should be explicitly referred in the main text. NB: RWGS in table is not explained.

Response: Thank you for your good comments as well as valuable suggestions in Q57-58. This is a very good suggestion. Table 1 information has been appropriately cited in the main context. The updated Table 1 is shown as below (next page):

Catalyst

Parameters

Conversion(%)

Selectivity(%)

T(℃)

SIE*

(J/mL)

Energy efficiency

  (mmol/kJ)

Remarks

Ref

CH4/ CO2 ratio

Catalyst loading (g)

Flow rate (mL/min)

Power (W)

CH4

CO2

H2

CO

LaNiO3@SiO2

1:1

0.2

40

150

88.31

77.76

83.65

92.43

200

225

0.17

Integrated with DBD, LaNiO3@SiO2 shows improved conversions and selectivity.

[86]

Ni/SiO2

1:1

0.2

50

86

26

16

47.4

52.9

110

103.2

N.A.

The combination of DBD and Ni/SiO2 catalyst enhance the activity of DRM due to the reaction between carbon-containing intermediates and oxygen radicals.

[60]

Ni/Al2O3

1:1

0.3

56

70

60

77

70-75

80

550

75

39%

The charge recombination on the Ni/Al2O3 catalyst surface will enhance the diffusion of carbon through the Ni catalyst and promote its oxidation by CO2.

[49]

Ni/Al2O3

4:1

6.4

50

1600

8.3

7.6

69

20

Ambient temperature

4.6 eV/molecule

4.5%

Ni addition enhanced the H2/CO ratio and reduced coke formation with the help of DBD.

[65]

Ni/γ-Al2O3

1:1

1.0

50

50

56.4

30.2

31

52.4

<150

60

0.32

The combination of plasma and Ni/γ-Al2O3 can increase the conversion rate of CH4.

[105]

Ni/ La2O3-MgAl2O4

1:1

0.5

20

100

86

84.5

50

49.5

350

300

0.13

La2O3 inhibits the RWGS* reaction, improves H2 selectivity and yield, and the formed intermediate carbonate (La2O2CO3) inhibits carbon deposition.

[55]

Ni/La2O3

1:1

0.2

50

160

63

54

71

85

150

240

0.14

Ni/LaO3 nanoparticles show excellent thermal stability in the DBD plasma reactor. La contributes to the formation of intermediates, which are responsible for activating CO2 and inhibiting carbon deposition.

[87]

Ni/ZrO2

1:1

0.6

50

200

53.57

60.81

82

95

650

240

N.A.

The Ni/ZrO2 catalyst prepared by the DBD plasma decomposition method greatly improves its activity due to its high dispersion and increased oxygen vacancies.

[56]

Ni-Co/ Al2O3-ZrO2

1:1

0.3

40

N.A.

58

62

95

100

850

N.A.

N.A.

The Ni-Co/Al2O3-ZrO2 catalyst after plasma treatment shows high catalytic activity due to its narrow particle size distribution, large surface area and strong metal-support interaction.

[70]

Ni-Mn/γ-Al2O3

1:1

0.5

30

2.2

28.4

13.2

23.2

40.5

N.A.

4.2

2.76

A higher activity and energy efficiency is achieved by the integrated plasma and Ni-Mn bimetallic catalyst system.

[20]

Ni-Mg/Al2O3

1.6:1

0.4

50

16

32

16

41.7

29.5

160

19.2

0.58

K promoted catalyst shows the best performance and enhances the energy efficiency of plasma process because it contains more active sites.

[63]

Ni-K/Al2O3

1.6:1

0.4

50

16

34

23

43.3

31.3

160

19.2

0.67

Ni-Ce/Al2O3

1.6:1

0.4

50

16

32

22

41.8

31.1

160

19.2

0.63

NiMgAlCe

4:6

0.45

90

48

36.1

22.5

N.A.

N.A.

N.A.

32

N.A.

Mg and Ce promoted the CO2 adsorption and increased the discharge area by partially tuning the filament discharge into surface discharge.

[79]

Mg,Ce-Ni/γ- Al2O3

1:1

N.A.

30

2.7

34.7

13

35

53.7

N.A.

5.4

1.97

Mobile oxygen and surface basicity effectively removed coke during methane activation by Ni and DBD.

[84]

*RWGS stands for the reverse water gas shift reaction; *SIE refers to specific input energy

Reviewer 3 Report

Comments to Authors

This paper contains many interesting facts about plasma-assisted catalytic dry reforming of methane based on Ni catalysts. It represents a comprehensive review covering 100 articles published from 2002 to this year. I think that the paper will be suitable for publication after being revised along with my comments.

However, as long as it is assumed that many people will read it, it is necessary that the statements and the terminology used should be correct.

 Page 1:  ..the synergistic effect of the catalyst and plasma improves the reaction efficiency.

 Page 6, line 281 The authors use the term: strong resistance to coking due to the synergetic effect [43-44].

Throughout the text many times: sometimes synergetic, sometimes synergistic.

By the way in the Conclusion is the mixture of these terms: “The synergetic effect of DBD plasma and different Ni catalysts are classified and discussed in two categories: pure Ni with various supports and Ni-based catalysts with doping. The synergistic effects of DBD plasma …”

Line 33: „Bob Dudly of BP stated…“  I have never heard about Bob Dudley; the proper citation should be provided.

Page 2, line 49: “appropriate for oxo and FTS reaction”; review paper – it should be understandable for a broad community of readers. What are oxo reactions?

Page 2, line 67: What is a luminous discharge? All mentioned discharges “such as gliding arc discharge, dielectric barrier discharge (DBD), luminous discharge, microwave discharge, corona discharge” (depending on the type of the gas) emit radiation.

Lines. 75-77: “In detail, the interaction between the catalyst and active species increases the life-time and collision probability of the reactive species, and enhances the surface modification and electric field [8]”.

I do not understand how the interaction between the catalyst and active species enhances the electric field. The electric field is probably non-uniform, so that can you kindly add a sentence explaining this effect? 

Page 2, line 93:  It is a macroscopic appearance composed of free electrons, ions (in which positive and negative charges are equal), and neutral particles, forming an electrically neutral non-condensing system [32-33].

Please look at the definition of a plasma:

“Positive and neutral charges are equal”: correctly: densities ne are approximately equal to ni.

Definition of plasma: Plasma is a quasineutral mixture of charged and neutral particles exhibiting collective behavior.

Or: …plasma is on average electrically neutral.

Page 3,

The SI unit of thermodynamic temperature is kelvin. (Alternatively can also be used the term degree of Celsius).  Correct unit of thermodynamic temperature is used on page 3, line 101: “between 5000 and 50000 K”. However, page line 127 is written: “usually thousands of degrees.”

Following part lines 131-140 must be reformulated entirely:

“The widely accepted and facile NTP generation method is to generate electricity by applying a high potential difference (breakdown voltage) between two electrodes. The potential can be alternating current (AC), direct current (DC) and pulse to provide electrical energy or the use of induction coils and microwaves. The intensity of the gas electric field is high enough to accelerate the electrons from one electrode to another, where very high energy can be developed. Influenced by the electric field shielding, a discharge is caused and the gas between the electrodes is converted into plasma [23,34]. The current common non-thermal plasma generation methods mainly include: dielectric barrier discharge, corona discharge, gliding arc discharge, glow discharge, microwave discharge and radio frequency discharge.

My remarks to this paragraph:

“The widely accepted and facile NTP generation method is to generate electricity by applying a high potential difference (breakdown voltage) between two electrodes.”

Do you mean that the NTP generation method is connected with the generation of electricity?

“The potential can be alternating current (AC) … “ please, look at the definition of potential and current.

“The intensity of the gas electric field is…“  Does it mean that the gas has an electric field? Is it something like the Earth has a gravitational field?

“…where very high energy can be developed” – please look at the conservation of energy law. (The law of conservation of energy states that energy can neither be created nor destroyed - only converted from one form to another).

“Influenced by the electric field shielding, a discharge is caused”.  I am sorry, but I do not understand how the electric field shielding causes a discharge.

“The current common non-thermal plasma generation methods mainly include: dielectric barrier discharge.”.

The method of plasma generation is something different than the type of discharges.

Page 4, line 151: “Compared with thermal plasma or low-temperature plasma, warm plasma has a higher…”  Can you compare the parameters of the thermal, low-temperature and warm plasma?

Besides, according to line 168: “Microwave discharge is warm plasma.”

I do not understand previous sentences.

In the explanation of microwave discharge, the authors use the term: non-polar discharge? What is it? I certainly understand that there exist polar and non-polar molecules.  However, I meet here for the first time with the term non-polar discharge.

Page 5. Formation of the corona discharge: “Its formation usually starts with high DC voltage applied to both ends of the electrode.”  Speaking frankly, I cannot imagine how to apply DC voltage only on one end of the conductive needle electrode or thin wire electrode. What happens if I apply DC voltage in the middle of the wire?

DBD is a gas power generation state in which discharge space is filled with an insulating medium.

The dielectric barrier discharge is a gas power generation state?

Can you explain what you mean by the “discharge space is filled with an insulating medium”.  Please explain based on Fig.1.

In this context please modify the text according to the reference 81- Kogelschatz - different types of DBD like volume or surface DBD.  

Page 6, line 252: Please complete the sentence: “Reduce the requirements for experimental equipment [40]”.

Page 6, line 274. “Although DRM shows a better performance in dielectric barrier power generation plasma reactors”. Can you please explain what are “dielectric barrier power generation plasma reactors”?

Page 6, line 287: “The main factors affecting the plasma performance are electric field enhancement and discharge.” Is it possible to specify in the text what is the plasma performance?

Page 13, lines 575-578: “The shape of the DBD plasma reactor is cylindrical. As shown in Figure 5a, it contains the following geometric parameters, which are discharge volume (VD, cm3), the discharge length (DL, cm), discharge gap (Dgap, mm) and material accumulation of the high-voltage electrode shape [8,81]”.

Please correct: Plasma reactor contains the following geometric parameters.

Can you explain what is: material accumulation of the high-voltage electrode shape?

Page 14, lines 612-613: “The dielectric material is very important to the structure of the reactor”. In my opinion, the dielectric material is mainly important for the mechanism of the discharge

Page 14, lines 629-931: Dgap can be expressed by using the difference between the radius of the dielectric and the high-voltage electrode.

This review is written for scientists working with the discharges and catalysts so that this explanation is not necessary. It is also seen in Fig.1 and Fig. 5b. However, if you wish to explain what is D gap, I suggest the following text: Dgap can be expressed by using the difference between the inner radius of the dielectric and the outer radius of the high-voltage electrode.

Is the discharge length identical with the length of the electrode? The discharge length is shown in Fig. 5b. What about the discharge length in Fig. 5c?

Page 14, line 654: “The discharge power can be adjusted by adjusting the voltage (V) and frequency (f).  As the voltage and frequency increase, the current pulse increases, thereby increasing the processing of the reactant gas.

In the case of the DBD there is only one current pulse?

Table 1: For the comparison, it will be useful to include a column giving the specific input energy into the table.  The fact is that the flowrate varies within a range from 56 to 20 (mL/min) and the power varies from 200 to 2.2 W. Besides, for Ni/Al2O3 is instead of power given the voltage 11 kV, so that comparison is not possible.

Conclusion

This review dealing with plasma-assisted catalytic dry reforming of methane based on Ni catalysts will be of interest to the scientific community working in the field of electrical discharges and various catalysts. Therefore, I recommend accepting the manuscript after minor revision, according to my comments.

Author Response

To Reviewer #3:

Comment: This paper contains many interesting facts about plasma-assisted catalytic dry reforming of methane based on Ni catalysts. It represents a comprehensive review covering 100 articles published from 2002 to this year. I think that the paper will be suitable for publication after being revised along with my comments.

This review dealing with plasma-assisted catalytic dry reforming of methane based on Ni catalysts will be of interest to the scientific community working in the field of electrical discharges and various catalysts. Therefore, I recommend accepting the manuscript after minor revision, according to my comments.

However, as long as it is assumed that many people will read it, it is necessary that the statements and the terminology used should be correct.

Response: Thank you for your positive comments and valuable suggestions to improve our manuscript. The responses are listed as below.

Q1: Page 1:  ..the synergistic effect of the catalyst and plasma improves the reaction efficiency.

 Page 6, line 281 The authors use the term: strong resistance to coking due to the synergetic effect [43-44].

Throughout the text many times: sometimes synergetic, sometimes synergistic.

By the way in the Conclusion is the mixture of these terms: “The synergetic effect of DBD plasma and different Ni catalysts are classified and discussed in two categories: pure Ni with various supports and Ni-based catalysts with doping. The synergistic effects of DBD plasma …”

Response: Thank you for your comments as well as valuable suggestions to improve our manuscript. We have corrected the terms and made it consistent at all the places. We have replaced ‘synergetic’ with ‘synergistic’ and highlighted them in red.

Q2: Line 33: „Bob Dudly of BP stated…“  I have never heard about Bob Dudley; the proper citation should be provided.

Response: Thank you for your valuable suggestions to improve our manuscript. The statement has been removed accordingly.

Q3: Page 2, line 49: “appropriate for oxo and FTS reaction”; review paper – it should be understandable for a broad community of readers. What are oxo reactions?

Response: Thank you for your good comments to improve our manuscript. The terms have been explained in the context accordingly and as below:

(3) The ratio of H2 to CO is close to 1, appropriate for oxo reaction (hydroformylation reaction in the presence of Co or Rh where olefins reacts with CO and H2 to produce aldehydes) and FTS reaction (also known as Fischer-Tropsch Synthesis is a process where CO and H2 reacts to form olefins and other valuable products).

Q4: Page 2, line 67: What is a luminous discharge? All mentioned discharges “such as gliding arc discharge, dielectric barrier discharge (DBD), luminous discharge, microwave discharge, corona discharge” (depending on the type of the gas) emit radiation.

Response: Thank you for your good comments to improve our manuscript. A non-thermal corona discharge is a weak luminous discharge which occurs at atmospheric pressure near sharp tips, edges or thin wires when the electric field is sufficiently large.

Q5: Lines. 75-77: “In detail, the interaction between the catalyst and active species increases the life-time and collision probability of the reactive species, and enhances the surface modification and electric field [8]”.

I do not understand how the interaction between the catalyst and active species enhances the electric field. The electric field is probably non-uniform, so that can you kindly add a sentence explaining this effect? 

Response: Thank you for your good comments to improve our manuscript. The sentences added are shown as below and highlighted in the context:

The electric field of plasma is enhanced due to charge accumulation and polarization effect caused by the roughness of catalysts. Also, a higher interaction between the catalyst and active species causes a higher conductivity which improves the magnitude of the electric field. Meanwhile, the catalyst facilitates the adsorption and prolongs the contact time of reactants, leading to a higher activation and conversion.

Q6: Page 2, line 93:  It is a macroscopic appearance composed of free electrons, ions (in which positive and negative charges are equal), and neutral particles, forming an electrically neutral non-condensing system [32-33].

Please look at the definition of a plasma:

“Positive and neutral charges are equal”: correctly: densities ne are approximately equal to ni.

Definition of plasma: Plasma is a quasineutral mixture of charged and neutral particles exhibiting collective behavior.

Or: …plasma is on average electrically neutral. 

Response: Thank you for your valuable comments to improve our manuscript. The authors have modified the definition of plasma according to the sources available in the literature and as mentioned by the reviewer.

It is a macroscopic appearance composed of free electrons, ions and neutral particles(in which positive and negative charges are equal), forming an electrically neutral non-condensing system [39-40].

Q7: Page 3,

The SI unit of thermodynamic temperature is kelvin. (Alternatively can also be used the term degree of Celsius).  Correct unit of thermodynamic temperature is used on page 3, line 101: “between 5000 and 50000 K”. However, page line 127 is written: “usually thousands of degrees.” 

Response: Thank you for your good comments to improve our manuscript. “thousands of degree” have been changed to “thousands of Kelvins”.

Q8: Following part lines 131-140 must be reformulated entirely:

“The widely accepted and facile NTP generation method is to generate electricity by applying a high potential difference (breakdown voltage) between two electrodes. The potential can be alternating current (AC), direct current (DC) and pulse to provide electrical energy or the use of induction coils and microwaves. The intensity of the gas electric field is high enough to accelerate the electrons from one electrode to another, where very high energy can be developedInfluenced by the electric field shielding, a discharge is caused and the gas between the electrodes is converted into plasma [23,34]. The current common non-thermal plasma generation methods mainly include: dielectric barrier discharge, corona discharge, gliding arc discharge, glow discharge, microwave discharge and radio frequency discharge.

Response: Thank you for your good comments to improve our manuscript. The paragraph has been revised as below:

In contract, the widely accepted and facile NTP generation method is by means of electricity. Capacitively coupled plasma is a typical example, where a large potential difference is exerted between two electrodes. Due to the electrical discharge generated by the electric field, the gas between the electrodes is transformed into plasma. Considering the thousands of volts voltage and around short distance between electrodes, the intensity of electric field in gas is high enough to accelerate the electrons from one electrode to another, where a very high energy can be transformed [2]. The current common non-thermal plasma generation methods mainly include: dielectric barrier discharge, corona discharge, gliding arc discharge, glow discharge, microwave discharge and radio frequency discharge.

Q9: “The widely accepted and facile NTP generation method is to generate electricity by applying a high potential difference (breakdown voltage) between two electrodes.”

Do you mean that the NTP generation method is connected with the generation of electricity?

Response: Thank you for your good comments to improve our manuscript. The statements have been revised as below:

In contract, the widely accepted and facile NTP generation method is by means of electricity. Capacitively coupled plasma is a typical example, where a large potential difference is exerted between two electrodes. Due to the electrical discharge generated by the electric field, the gas between the electrodes is transformed into plasma.

Q10: “The potential can be alternating current (AC) … “ please, look at the definition of potential and current.

Response: Thank you for your good comments to improve our manuscript. The statements have been removed.

Q11: “The intensity of the gas electric field is…“  Does it mean that the gas has an electric field? Is it something like the Earth has a gravitational field?

Response: Thank you for your good comments to improve our manuscript. The statements have been revised as below:

Considering the thousands of volts voltage and short distance between electrodes, the intensity of electric field in gas is high enough to accelerate the electrons from one electrode to another, where a very high energy can be transformed [2].

Q12: “…where very high energy can be developed” – please look at the conservation of energy law. (The law of conservation of energy states that energy can neither be created nor destroyed - only converted from one form to another).

Response: Thank you for your good comments to improve our manuscript. “developed” is changed to “transformed”.

Q13: “Influenced by the electric field shielding, a discharge is caused”.  I am sorry, but I do not understand how the electric field shielding causes a discharge.

Response: Thank you for your good comments to improve our manuscript. The statement has been removed.

Q14: “The current common non-thermal plasma generation methods mainly include: dielectric barrier discharge.”.

The method of plasma generation is something different than the type of discharges.

Response: Thank you for your good comments to improve our manuscript. The statements have been changed to “The current common non-thermal plasma mainly include”.

Q15: Page 4, line 151: “Compared with thermal plasma or low-temperature plasma, warm plasma has a higher…”  Can you compare the parameters of the thermal, low-temperature and warm plasma?

Besides, according to line 168: “Microwave discharge is warm plasma.”

I do not understand previous sentences.

In the explanation of microwave discharge, the authors use the term: non-polar discharge? What is it? I certainly understand that there exist polar and non-polar molecules.  However, I meet here for the first time with the term non-polar discharge.

Page 5. Formation of the corona discharge: “Its formation usually starts with high DC voltage applied to both ends of the electrode.”  Speaking frankly, I cannot imagine how to apply DC voltage only on one end of the conductive needle electrode or thin wire electrode. What happens if I apply DC voltage in the middle of the wire?

Response: Thank you for your good comments to improve our manuscript. The information about the other type of discharge has been removed according to the other reviewer’s comment.

Q16: DBD is a gas power generation state in which discharge space is filled with an insulating medium.

The dielectric barrier discharge is a gas power generation state?

Can you explain what you mean by the “discharge space is filled with an insulating medium”.  Please explain based on Fig.1.

In this context please modify the text according to the reference 81- Kogelschatz - different types of DBD like volume or surface DBD.

Response: Thank you for your valuable comments to improve our manuscript. The revised sentences for this part are shown as below:

DBD is defined in the scenario where the dielectric is placed between two electrodes, and the discharge space is filled with an insulating medium.

The discharge space is usually filled with reacting gas or vacuum.

Q17: Page 6, line 252: Please complete the sentence: “Reduce the requirements for experimental equipment [40]”.

Response: Thank you for your valuable comments to improve our manuscript. The statement has been removed as it doesn’t add any meaning to the previous sentences.

Q18: Page 6, line 274. “Although DRM shows a better performance in dielectric barrier power generation plasma reactors”. Can you please explain what are “dielectric barrier power generation plasma reactors”?

Response: Thank you for your valuable comments to improve our manuscript. The authors realized that this sentence has been written by mistake. It is better to write dielectric barrier plasma reactors, which is corrected in the revised manuscript.

Q19: Page 6, line 287: “The main factors affecting the plasma performance are electric field enhancement and discharge.” Is it possible to specify in the text what is the plasma performance?

Response: Thank you for your valuable comments to improve our manuscript. The plasma performance can be measured in terms of output voltage and current intensity. The statement is added just after the plasma performance.

Q20: Page 13, lines 575-578: “The shape of the DBD plasma reactor is cylindrical. As shown in Figure 5a, it contains the following geometric parameters, which are discharge volume (VD, cm3), the discharge length (DL, cm), discharge gap (Dgap, mm) and material accumulation of the high-voltage electrode shape [8,81]”.

Please correct: Plasma reactor contains the following geometric parameters.

Response: Thank you for your valuable comments to improve our manuscript. The revised statement has been added accordingly: the plasma reactor contains the following geometric parameters.

Q21: Can you explain what is: material accumulation of the high-voltage electrode shape?

Response: Thank you for your valuable comments to improve our manuscript. Herein, the authors would like to make the correction for the sentence.

Current: ‘and material accumulation of the high-voltage electrode shape’.

Correction: ‘shape and material of the high-voltage electrode.’

Q22: Page 14, lines 612-613: “The dielectric material is very important to the structure of the reactor”. In my opinion, the dielectric material is mainly important for the mechanism of the discharge

Response: Thank you for your valuable comments to improve our manuscript. Corrections according to the reviewer’s comment have been made as below:

The dielectric material is mainly important for the mechanism of the discharge.

Q23: Page 14, lines 629-931: Dgap can be expressed by using the difference between the radius of the dielectric and the high-voltage electrode.

Response: Thank you for your valuable comments to improve our manuscript. The authors have made the changes according to the reviewer’s suggestion as below:

Dgap can be expressed by using the difference between the inner radius of the dielectric and the outer radius of the high-voltage electrode.

Q24: Is the discharge length identical with the length of the electrode? The discharge length is shown in Fig. 5b. What about the discharge length in Fig. 5c?

Response: Thank you for your valuable comments to improve our manuscript. The discharge length is not identical to the length of electrode. It is length considered inside the electric field or the length inside the discharge zone.

Q25: Page 14, line 654: “The discharge power can be adjusted by adjusting the voltage (V) and frequency (f).  As the voltage and frequency increase, the current pulse increases, thereby increasing the processing of the reactant gas.

In the case of the DBD there is only one current pulse?

Response: Thank you for your valuable comments to improve our manuscript. The current pulse can be changed by changing voltage and frequency.

Q26: Table 1: For the comparison, it will be useful to include a column giving the specific input energy into the table. The fact is that the flowrate varies within a range from 56 to 20 (mL/min) and the power varies from 200 to 2.2 W. Besides, for Ni/Al2O3 is instead of power given the voltage 11 kV, so that comparison is not possible.

Response: Thank you for your valuable comments to improve our manuscript. The Table 1 is modified and added in the revised manuscript and as below (next page):

Catalyst

Parameters

Conversion(%)

Selectivity(%)

T(℃)

SIE*

(J/mL)

Energy efficiency

  (mmol/kJ)

Remarks

Ref

CH4/ CO2 ratio

Catalyst loading (g)

Flow rate (mL/min)

Power (W)

CH4

CO2

H2

CO

LaNiO3@SiO2

1:1

0.2

40

150

88.31

77.76

83.65

92.43

200

225

0.17

Integrated with DBD, LaNiO3@SiO2 shows improved conversions and selectivity.

[86]

Ni/SiO2

1:1

0.2

50

86

26

16

47.4

52.9

110

103.2

N.A.

The combination of DBD and Ni/SiO2 catalyst enhance the activity of DRM due to the reaction between carbon-containing intermediates and oxygen radicals.

[60]

Ni/Al2O3

1:1

0.3

56

70

60

77

70-75

80

550

75

39%

The charge recombination on the Ni/Al2O3 catalyst surface will enhance the diffusion of carbon through the Ni catalyst and promote its oxidation by CO2.

[49]

Ni/Al2O3

4:1

6.4

50

1600

8.3

7.6

69

20

Ambient temperature

4.6 eV/molecule

4.5%

Ni addition enhanced the H2/CO ratio and reduced coke formation with the help of DBD.

[65]

Ni/γ-Al2O3

1:1

1.0

50

50

56.4

30.2

31

52.4

<150

60

0.32

The combination of plasma and Ni/γ-Al2O3 can increase the conversion rate of CH4.

[105]

Ni/ La2O3-MgAl2O4

1:1

0.5

20

100

86

84.5

50

49.5

350

300

0.13

La2O3 inhibits the RWGS* reaction, improves H2 selectivity and yield, and the formed intermediate carbonate (La2O2CO3) inhibits carbon deposition.

[55]

Ni/La2O3

1:1

0.2

50

160

63

54

71

85

150

240

0.14

Ni/LaO3 nanoparticles show excellent thermal stability in the DBD plasma reactor. La contributes to the formation of intermediates, which are responsible for activating CO2 and inhibiting carbon deposition.

[87]

Ni/ZrO2

1:1

0.6

50

200

53.57

60.81

82

95

650

240

N.A.

The Ni/ZrO2 catalyst prepared by the DBD plasma decomposition method greatly improves its activity due to its high dispersion and increased oxygen vacancies.

[56]

Ni-Co/ Al2O3-ZrO2

1:1

0.3

40

N.A.

58

62

95

100

850

N.A.

N.A.

The Ni-Co/Al2O3-ZrO2 catalyst after plasma treatment shows high catalytic activity due to its narrow particle size distribution, large surface area and strong metal-support interaction.

[70]

Ni-Mn/γ-Al2O3

1:1

0.5

30

2.2

28.4

13.2

23.2

40.5

N.A.

4.2

2.76

A higher activity and energy efficiency is achieved by the integrated plasma and Ni-Mn bimetallic catalyst system.

[20]

Ni-Mg/Al2O3

1.6:1

0.4

50

16

32

16

41.7

29.5

160

19.2

0.58

K promoted catalyst shows the best performance and enhances the energy efficiency of plasma process because it contains more active sites.

[63]

Ni-K/Al2O3

1.6:1

0.4

50

16

34

23

43.3

31.3

160

19.2

0.67

Ni-Ce/Al2O3

1.6:1

0.4

50

16

32

22

41.8

31.1

160

19.2

0.63

NiMgAlCe

4:6

0.45

90

48

36.1

22.5

N.A.

N.A.

N.A.

32

N.A.

Mg and Ce promoted the CO2 adsorption and increased the discharge area by partially tuning the filament discharge into surface discharge.

[79]

Mg,Ce-Ni/γ- Al2O3

1:1

N.A.

30

2.7

34.7

13

35

53.7

N.A.

5.4

1.97

Mobile oxygen and surface basicity effectively removed coke during methane activation by Ni and DBD.

[84]

*RWGS stands for the reverse water gas shift reaction; *SIE refers to specific input energy.

Round 2

Reviewer 2 Report

i would like to thank he authors for a details revision of their manuscirpt.  they have done a substantial works. i am glad i was able to help them to improve the quality of the manuscript.

Author Response

Dear Editor Ms. Maeve Yue,

Thank you very much for you and the referees’ efforts to the review of our manuscript entitled “Recent developments in plasma-assisted catalytic dry reforming of methane based on Ni catalysts”. We acknowledge your and the reviewers’ comments and constructive suggestions very much, which are valuable for improving the quality of our manuscript. After a careful and serious consideration of the comments, we revised the manuscript as you and referees suggested. Therefore, we sincerely hope you could consider our manuscript for publication in Catalysts.

Yours sincerely

  1. Kawi (Ph.D.)

Associate Professor

Department of Chemical & Biomolecular Engineering

National University of Singapore

Singapore

Responses to the editor and reviewers

We thank you and reviewers for your careful review of our manuscript, and really appreciate your constructive comments. Note that all the changes/additions are highlighted in red color in the revised version of the manuscript. Please see below for our detailed responses to the comments.

To Editor:

Q1: Please carefully check the accuracy of names and affiliations.

Response: Thank you. Names and affiliations are correct.

Q2: Please add phone number.

Response: Thank you. Phone number is added.

Q3: Please change “x” to “×” and add space before and after “+”.

Response: Thank you. Revised figure is updated and shown as below:

Q4: Please add explanations for a,b,c,d in figure caption.

Response: Thank you. Explanations have been added in the figure remarks and below:

(a) Pre-packing fixed-bed DBD reactor; (b) In situ packing fixed-bed DBD reactor; (c) Post packing fixed-bed DBD reactor; (d) Fully packed fixed-bed DBD reactor.

Q5: Please check if individual contribution for each co-author has been stated.

Response: Thank you. Yes, the contribution has been stated.

Q6: In this section, please provide details regarding where data supporting reported results can be found, including links to publicly archived datasets analyzed or generated during the study. Please refer to suggested Data Availability Statements in section “MDPI Research Data Policies” at https://www.mdpi.com/ethics. You might choose to exclude this statement if the study did not report any data.

Response: Thank you. All data included in this study are available upon the permission from the publishers or cited.

Q7: In this section, you can acknowledge any support given which is not covered by the author contribution or funding sections. This may include administrative and technical support, or donations in kind (e.g., materials used for experiments).

Response: Thank you. We would like to leave this part empty.

To Reviewer #2:

Comment: I would like to thank he authors for a details revision of their manuscript. They have done a substantial works. I am glad I was able to help them to improve the quality of the manuscript.

Response: Thank you for your positive comments and valuable suggestions to improve our manuscript.
